# Safe drugs with high potential to block malaria transmission revealed by a spleen-mimetic screening

Mario Carucci [1,14], Julien Duez [2,14], Joel Tarning [3,4], Irene García-Barbazán [5], Aurélie Fricot-Monsinjon[1], Abdoulaye Sissoko [1], Lucie Dumas[1], Pablo Gamallo [6], Babette Beher[1], Pascal Amireault [1,7], Michael Dussiot[7,8], Ming Dao [9], Mitchell V. Hull[10], Case W. McNamara[10], Camille Roussel[1,8,11], Papa Alioune Ndour [1], Laura Maria Sanz [6], Francisco Javier Gamo [6] & Pierre Buffet[1,12,13] ✉

Malaria parasites like *Plasmodium falciparum* multiply in red blood cells (RBC), which are cleared from the bloodstream by the spleen when their deformability is altered. Drug-induced stiffening of *Plasmodium falciparum*-infected RBC should therefore induce their elimination from the bloodstream. Here, based on this original mechanical approach, we identify safe drugs with strong potential to block the malaria transmission. By screening 13 555 compounds with spleen-mimetic microfilters, we identified 82 that target circulating transmissible form of *P. falciparum*. NITD609, an orally administered PfATPase inhibitor with known effects on *P. falciparum*, killed and stiffened transmission stages in vitro at nanomolar concentrations. Short exposures to TD-6450, an orally-administered NS5A hepatitis C virus inhibitor, stiffened transmission parasite stages and killed asexual stages in vitro at high nanomolar concentrations. A Phase 1 study in humans with a primary safety outcome and a secondary pharmacokinetics outcome (https://clinicaltrials.gov, ID: NCT02022306) showed no severe adverse events either with single or multiple doses. Pharmacokinetic modelling showed that these concentrations can be reached in the plasma of subjects receiving short courses of TD-6450. This physiologically relevant screen identified multiple mechanisms of action, and safe drugs with strong potential as malaria transmission-blocking agents which could be rapidly tested in clinical trials.

Between 2000 and 2015, malaria incidence and mortality decreased by 27% and 50%, respectively. The decline in incidence stopped between 2015 and 2019, and in 2020 there was even a worrying re-increase of both incidence and mortality; with 241 million of malaria-related cases and 627,000 deaths[1]. The emergence of artemisinin-resistant *P. falciparum* strains in South-East Asia[2–4] and recently in Africa[5,6] further threatens the control of the disease. Blocking the transmission of *P.*

*falciparum* from its human host to its mosquito vector is therefore envisioned to reduce malaria incidence and hopefully contribute to its global eradication[7]. Stage V gametocytes, the mature sexual stages of malaria parasites, are the only form transmitted to the *Anopheles* species vector and their low number creates a natural bottleneck in the parasite life cycle, making them a fine target to block parasite transmission[8].

The spleen retains stiff erythrocytes and clears them from the circulation[9,10]. Any one erythrocyte will spend no more than two hours in the circulation before being squeezed through narrow inter-endothelial slits in the walls of the splenic sinuses[11,12]. In malaria, only the most deformable stages, namely rings (immature asexual stage) and stage V gametocytes, can overcome this mechanical challenge and persist in the circulation[8,13–16]. The switch from stiff immature to deformable mature gametocytes occurs at the end of their 10 to 12-day maturation process, between stages IV and V[13,16]. Drug-induced stiffening of stage V gametocytes should therefore remove them from the transmission cycle. Several screening approaches to identify gametocyte-targeted compounds able to kill or inactivate the parasite have been explored[17–22]. These approaches have identified interesting candidate targets displaying variable druggability. Recently, large libraries of drugs amenable to repurposing have been made available[23]. This offers an opportunity to identify effective and rapidly deployable transmission-blocking candidates[24]. With the aim of finding parasite targets, we have established a physiologically relevant approach to screen thousands of compounds for their stiffening activity on parasitized erythrocytes.

Our approach uses a splenic mimetic filtration method, called microsphiltration, that quantifies the ability of erythrocytes to squeeze between microspheres[25]. This method has been adapted to drug-screening[26–28] then combined with the assessment of parasite killing based on active mitochondrial staining[19]. For the present work, we used this bio-mimetic approach to screen more than 12,000 compounds from a repurposing library for their ability to induce the retention of mature *P. falciparum* gametocytes in the human spleen.

## Results

### Dual screening of 3 libraries identifies 119 primary hits with stiffening activity or killing effect on gametocytes

Screening was conducted as previously described[27]. Briefly, mature stage V gametocytes (NF54 strain) were cultured in vitro, then exposed to the explored compounds and filtered thereafter through layers of microspheres (microsphiltration)[25,28]. Stiffening activity was determined by comparing gametocytemia up and downstream from the microsphere layers after filtration of drug-exposed gametocytes in 384-well plates using a vacuum manifold. Gametocytemia was assessed using DNA (Sybr Green) and erythrocyte membrane (CellMask) staining. Killing effect was assessed by MitoTracker staining[19]. We screened two small libraries, specifically the Pathogen Box from the Medicine for Malaria Venture and the Kinase Inhibitors Box from GSK (400 and 350 compounds, respectively, Fig. 1A, B), and furthermore the larger ReFrame repurposing library (12,805 compounds, Fig. 1C).

Primary screening was repeated six or three times for small libraries (Kinase Inhibitors and Pathogen Boxes). Hits were selected plate by plate, based on linear regression of compound distributions on screening results plots (Fig. 2C, D). We found three and four hits respectively in the Pathogen and Kinase Inhibitors Boxes (Fig. 1A, B). Their respective hit rates were 0.75%, and 1.14%. Compounds active in some but not all screening replicates were added to the hits for further analysis. The larger ReFrame library was screened in singlicate, raising 112 hits (0.87% hit rate, Fig. 1C). The detected hits had either a predominant stiffening activity (44%), a predominant killing effect (28%), or a combination of both (28%, Fig. 2A, B). Z' values were between 0.4 and 0.7 (Fig. S1). These 112 hits, plus 63 compounds (0.6%) with uninterpretable results upon primary screening were explored further.

### Dose-response analysis confirms the activity of 82 hits, including candidates from previous screening campaigns

119 hits were selected for dose-response analysis (DRA): three from the Pathogen Box, four from the Kinase Inhibitors Box, and 112 from the ReFrame library. Confirmation of the stiffening effect, killing effect or coexistence of both was the criterion for further analysis of hits. $IC_{50}$

values were obtained for both killing effect and stiffening activity (Fig. 3, Table S1) but no $IC_{50}$ cut-off value was used for further hit prioritization. Hits were deprioritized when no $IC_{50}$ could be determined. Six of the seven hits from the Pathogen and Kinase Inhibitors Boxes were confirmed (86% confirmation rate, Fig. 1A, B). A further 17 compounds (12 from the Pathogen Box and 5 from the Kinase Inhibitors Box) that showed effects in some but not all screening replicates were also selected for DRA. None of them were confirmed as active.

The three DRA-confirmed hits from the Pathogen Box, namely MMV020081, MMV667494 and MMV030734 had been identified in previous gametocyte-targeting screenings of this library[29–31]. Two of them, MMV667494 and MMV030734, showed high killing activity specific to female gamete formation (Table S2). MMV667494 and MMV030734 induced gametocyte stiffening at concentrations lower than those inducing killing. All confirmed hits from the Kinase Inhibitors Box had the same chemical scaffold (Table S2). One chemical analog, not tested in primary screening, was added to this list and tested in DRA. This compound showed $IC_{50}$ values close to those of the most powerful hits (2.44 and 0.8 μM for killing and retention, respectively, Table S2) and was used for further post-screening validation. Confirmed hits from the ReFrame library had either a predominant stiffening activity (40% of hits), a predominant killing effect (30%) or a combination of both (30%) (Table S1).

Of the 112 hits from the ReFrame library, 74 were confirmed by DRA (66% confirmation rate, Fig. 1C). An additional 63 compounds with uninterpretable results upon primary screening were also selected for DRA. For 15 of these 63 compounds, the corresponding well of the 384-well microsphiltration plate was not operational due to leakage of microspheres, a rare event[27]. For 46 compounds, the microscope failed to capture at least one readable image. Finally, we tested sildenafil and tadalafil by DRA. Although these 2 drugs were not captured by the primary screen they have been previously reported to induce stiffening of stage V gametocytes[32,33]. $IC_{50}$ values were obtained for both killing effect and stiffening activity or, in some cases, one of them (Fig. 3, Table S1). We found two confirmed hits among those 63 compounds, which is in line with the hit rate of the whole screening campaign (0.87%). These two hits were L-tetrahydropalmatine (L-THP) and Zinc Pyrithione.

### Confirmed hits fall into seven major groups

Structures, targets and results of previous screening approaches regarding confirmed hits from the Pathogen Box and the Kinase Inhibitors Box are indicated in the Table S2. Hits from the ReFrame library fell into seven groups (Fig. 1 and Table S1) based on known molecular targets in human subjects or medical indications: histone deacetylase (HDAC) inhibitors (9 hits); cardiac glycosides targeting the sodium-potassium ATPase pump (7 hits); kinase and phosphatase inhibitors (6 hits); monoamine oxidase (MAO) inhibitors like the known malaria transmission-blocking candidate methylene blue (2 hits)[34] also selected by previous gametocytes screening campaigns[35,36]; antimalarial agents (6 hits); antibiotic/antiviral agents (18 hits) and a final group of compounds not allocatable to a defined group (28 hits). The known molecular targets of the hits in human cells, bacteria, viruses or parasites were listed (Table S1). Analysis of homologies in *P. falciparum* may identify pathways for gametocytes killing and/or stiffening yet to be characterized.

### Hit profiling identifies a safe and orally-administered NS5A inhibitor not previously known for its antimalarial activity

Hits from the ReFrame library were further selected based on safety, preferably oral administration (high score if oral), and potential availability for widespread use. Briefly, selected hits that were not orally administered in animal models or in human subjects were excluded (44 confirmed hits excluded out of 76). Then, the remaining 32 hits were ranked for safety and PK. As anti-transmission imposes

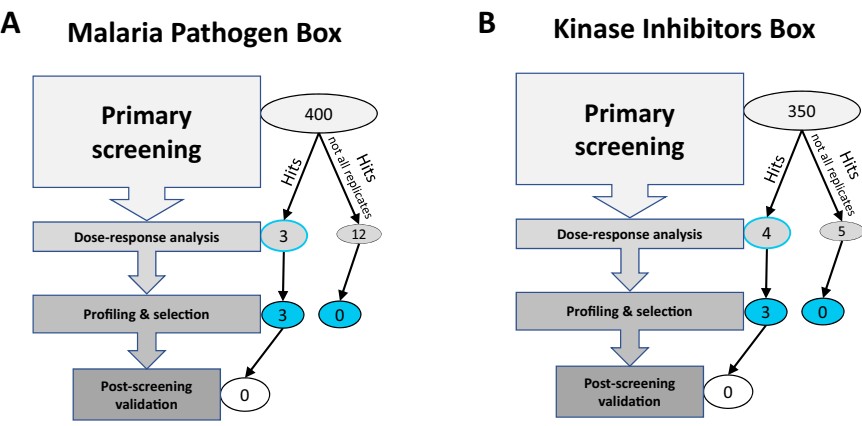

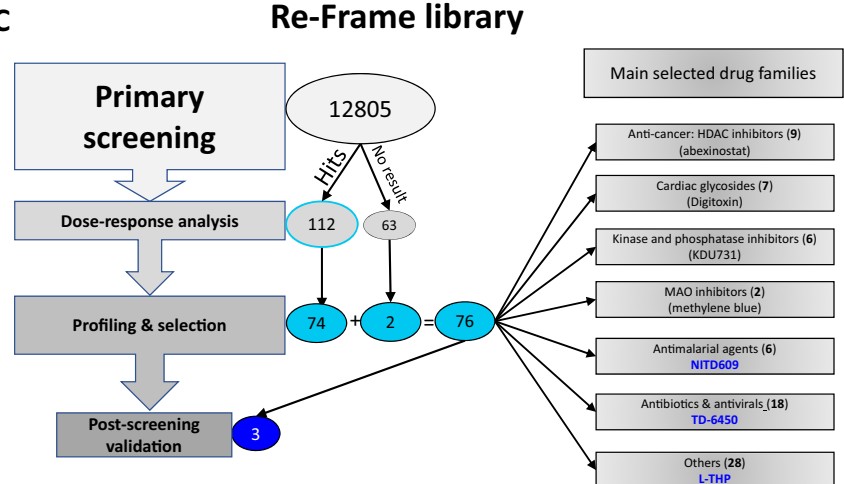

**Fig. 1 | High-throughput screening based on mitochondrial staining and cell deformability identifies compounds with both killing effect and stiffening activity on *P. falciparum* late gametocytes.** Screening progression cascade of three different libraries: Malaria Pathogen Box (**A**), Kinase Inhibitors Box (**B**), and ReFrame library (**C**). **A** 3 hits from primary screening were submitted to dose-response analysis along with 12 compounds found active in some but not all screening replicates. The 3 hits were confirmed but none of them was selected for further post-screening validation. **B** Four hits from primary screening along with 5 compounds found active in some but not all screening replicates were submitted to dose-response analysis raising 3 confirmed hits. None of them was selected for further post-screening validation. **C** 112 hits from primary screening were submitted to dose-response analysis, raising 74 confirmed hits. 63 compounds with uninterpretable results during primary screening were added to the hits for dose-response analysis raising additional two confirmed hits. The 76 confirmed hits were allocated to seven groups (panels on the right), based on their activity and molecular target. For each group, one representative hit has been selected for illustration. Hit scoring based on route of administration, safety in human subjects, and pharmacokinetics resulted in the selection of 3 drugs submitted to final confirmation experiments (dark blue).

almost perfect safety, drugs that showed serious adverse events were excluded. Finally, hits with the best therapeutic window (serum peak greater or close to IC$_{50}$) were explored further. For example, MMV-390048, an antimalarial drug with a good safety profile, was excluded from further analysis because of an absent or very narrow therapeutic window (IC$_{50}$ 3.5-3.9 μM, serum peak concentration 2.8 μM[37]). The three most attractive drugs according to this selection were NITD609, L-THP and TD-6450 (Table 1). NITD609, also known as cipargamin, is a powerful inhibitor of *P. falciparum* ATPase 4. It displays a strong killing effect at nanomolar concentrations, both on asexual and sexual parasite stages[38,39] and a stiffening activity on asexual parasites[40]. NITD609 is currently being tested in a large multi-site Phase 2 trial in Africa. The present results confirmed its known effect on gametocytes[41] but furthermore revealed its stiffening activity on mature gametocytes (Figs. 2C and 3A), which is expected to induce very fast clearance of transmittable forms present in the circulation at the time of treatment. L-THP is an alkaloid with anxiolytic and sedative effects currently under development in the setting of psychiatric disorders and addictive behaviors[42]. However, it also has a known effect

on asexual stages of *P. falciparum*[43]. Analyses in the present expanded its antimalarial spectrum to the parasite's sexual stages (Fig. 3C). TD-6450 is a NS5A inhibitor developed for the treatment of Hepatitis C virus infection. Despite solid indications of excellent safety (Table 2) and anti-HCV efficacy similar to those of other members of this drug family, the development of TD-6450 as a therapy for hepatitis C was stopped after Phase 2 for strategic reasons. L-THP was not explored further because, despite its known activity of asexual stages[43], its stiffening activity was not fully confirmed upon post-screening. TD-6450 and NITD609 were retained for further explorations in the present study.

### TD-6450 stiffens gametocytes and kills asexual stages of *P. falciparum* at sub-micromolar to low micromolar concentrations

The stiffening activities of TD-6450 and two control compounds, namely NITD609 and GSK1321730A, were confirmed by DRA with a parasite strain (3D7) other than that used during screening (NF54). This confirmation was performed in a different laboratory (Paris versus Madrid) using a different microplate format (96 wells in Paris v. 384

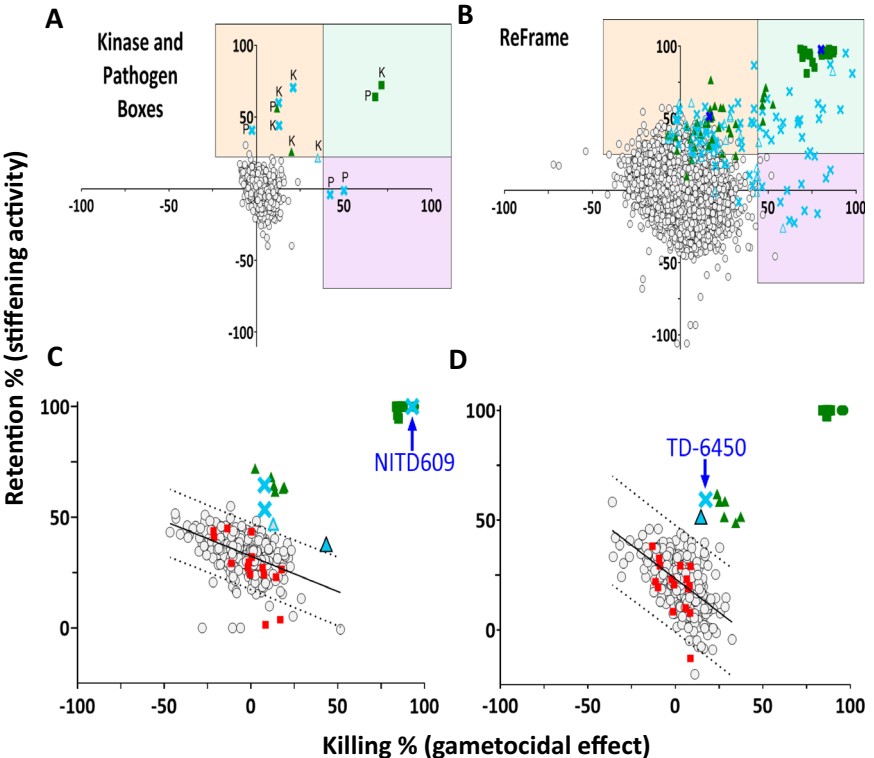

**Fig. 2 | Compiled screening results and representative plate analysis.** Representative screening output scatterplots for the specific (**A**) and ReFrame (**B**) libraries, and for 2 single plates from the ReFrame primary screening, with NITD609 (**C**) and TD-6450 (**D**) as finally selected hits. In panel A, "K" refers to Kinase Inhibitors Box and "P" refers to Pathogen Box. Killing and retention rates are on the x and y axes, respectively. Negative controls (red squares), and positive controls including calyculin A 75 nM for stiffening activity (green triangles), Gentian violet 50 μM for killing effect (green circles) and NITD609 0.5 μM for combined stiffening and killing effect (green squares), fell inside and outside of the main cloud of inactive test compounds (empty gray circles) defined by linear regression +/−95% prediction bands SD (full line and dotted lines, respectively). Hits were either confirmed by DRA (Blue X), or not (blue empty triangles). Source data are provided as a Source Data file.

wells in Madrid). TD-6450 median (±SD) IC$_{50}$s were of sub-micromolar range (0.28 ± 0.58 μM, five independent experiments). Normalized retention rates showed significant effects for both NITD609 (66%, $p < 0.0001$), and TD-6450 (27% $p < 0.0001$, Fig. 4B), with a higher stiffening activity for NITD609 compared to TD-6450. Combining NITD609 and TD-6450 significantly enhanced gametocyte stiffening compared to either drug used alone, i.e. 73% versus 66% for NITD609 ($p = 0.0058$) and versus 27% for TD-6450 ($p < 0.0001$) (Fig. 4B). The sustainability of drug-induced stiffening was also analyzed. Gametocytes were cultured with exposure to the drugs for 24 hours and retentions remeasured by microsphiltration an additional 24 h after drug washout (Fig. 4C). In this setting, TD-6450 showed a better sustained stiffening activity than NITD609 did (43% versus 23%, $p = 0.0525$) (Fig. 4C). Microscopic observations confirmed the known parasite-swelling effect mediated by NITD609 while no morphological change was observed in gametocytes exposed to TD-6450[44]. This observation was confirmed by aspect ratio values obtained by imaging flow cytometry (Fig. 4A). The IC$_{50}$s of TD-6450 during the screening campaign (in Madrid) were either 455 ± 304 nM or 2.03 ± 0.35 μM when respectively all gametocytes or only females were stained (2 experiments). These readout-dependent differences suggest that TD-6450 is predominantly active on male gametocytes as previously observed with several anti-gametocyte compounds[18], possibly explaining the apparent partial effect with this drug.

Concerning the efficacy of TD-6450 on asexual parasites, IC$_{50}$ was 875 ± 205 nM on 3D7 and 1.33 ± 0.23 μM on NF54 strains (2 experiments, Fig. 5B). Microscopic observation on Giemsa-stained smears confirmed the swelling of NITD609-exposed rings, while TD-6450-exposed asexual rings showed delayed maturation and pyknosis without swelling (Fig. 5A

and Fig. S2), suggesting that these drugs operate through different mechanisms. To assess the ability of *P. falciparum* to develop resistance against TD-6450, we exposed F32-TEM line parasites, a reference artemisinin-sensitive *P. falciparum* strain[45], to the drug at 3 μM concentration for three days. We confirmed the absence of parasites on Giemsa-stained smears following exposure, then checked daily for parasite re-appearance in a drug-free culture flask. We observed a 3.5- and a 1.5-fold shift in the IC$_{50}$ of TD-6450 on parasites[45] exposed to five or ten such drug-killing pulses, respectively (Fig. 5C), without shortening of time-to-regrowth from the first to the last pulse. This low level of IC$_{50}$ shift suggests that there is no rapid emergence of parasites markedly resistant to TD-6450 in vitro.

### Pharmacokinetics of TD-6450
TD-6450 was discovered and developed for the treatment of Hepatitis C virus infection by Theravance Biopharma Inc, but was stopped after phase II for strategic reasons. The first phase I clinical trial was completed in 2014 (clinical trials ID: NCT02022306). The primary objective of this study was to evaluate the safety and tolerability of single ascending dose (SAD) and multiple ascending dose (MAD) in healthy subjects (Table 2). The secondary objective was to determine the pharmacokinetics (PK) of TD-6450 in healthy subjects for both SAD and MAD. The PK data obtained from this phase I study were used to make a concentration-time modeling. Goodness-of-fit was high (Fig. 6A) with data from the single ascending dose (SAD) and multiple ascending dose (MAD) studies performed in healthy volunteers (Fig. 6B, C). The following additional covariate relationships were retained in the final model: dose on relative bioavailability, implemented as a hockey-stick function resulting in a linear

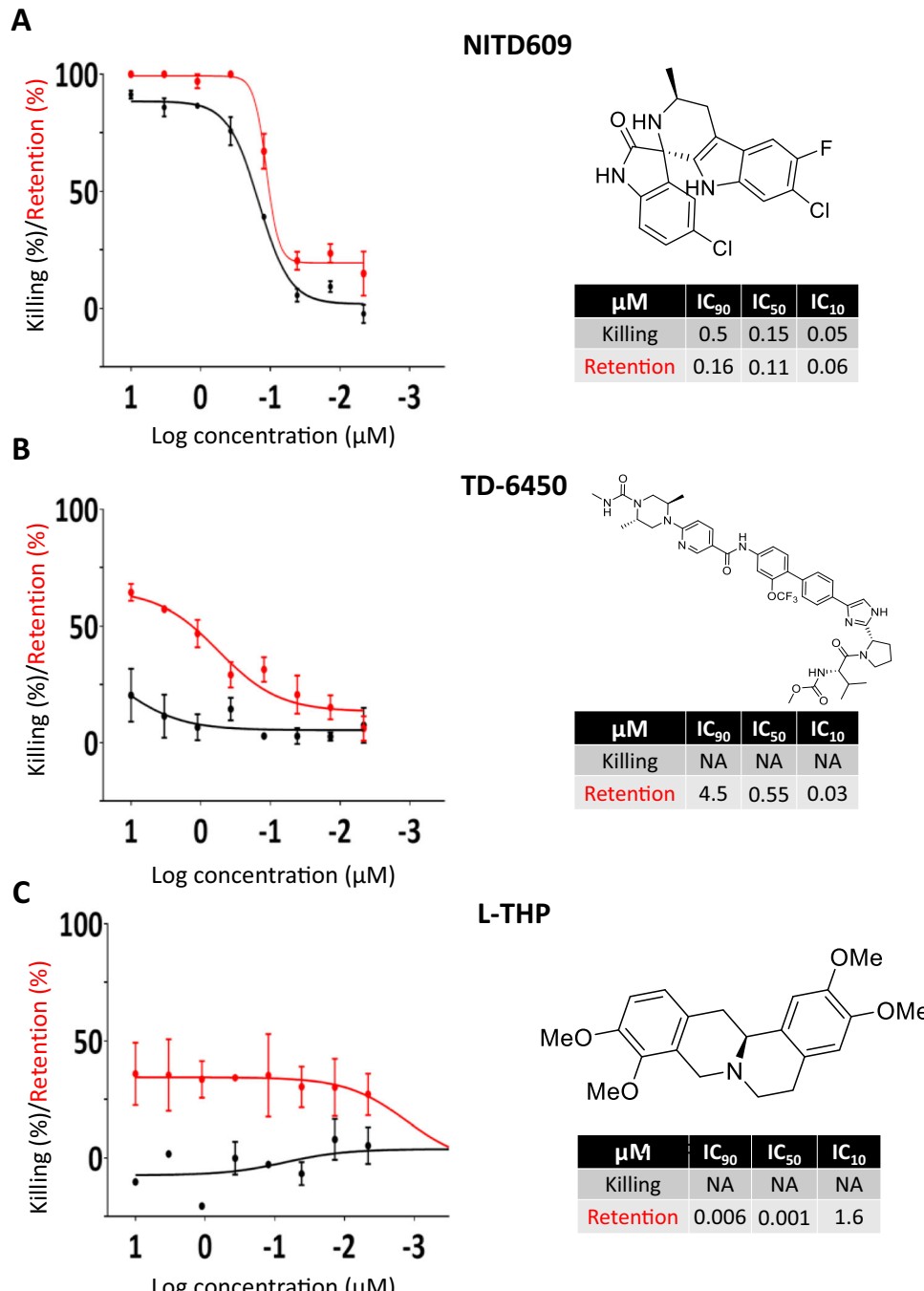

**Fig. 3 | Dose-response curves of selected hits.** Dose-response curves and chemical structures of the 3 selected hits: NITD609 (**A**), TD-6450 (**B**) and L-THP (**C**). TD-6450 was a pure enantiomer. Stereocenters are: (S)−1-((S)−2-(5-(4'-(6-((2 R,5 S). Data are presented as mean values +/− SEM. For NITD609 and L-THP, $n = 2$ independent experiments, for TD-6450, $n = 4$ independent experiments. Source data are provided as a Source Data file.

decrease in relative bioavailability from doses of 240 mg to 500 mg (i.e. 24.7% decrease per 100 mg increase in dose, above 240 mg); dose on elimination clearance implemented as a saturation function reaching full saturation at the higher dose; duration of treatment on relative bioavailability implemented as a saturation function reaching full saturation at steady-state dosing; food intake on relative bioavailability, implemented as a categorical covariate effect (i.e., 68.5% higher bioavailability when administered with food); and food intake on absorption rate implemented as a categorical covariate effect (i.e. 93.3% longer absorption time when administered with food) (Table 3).

Based on the modeling output, concentration-time simulations (Fig. S3) of 1000 mg single dose without (Fig. 7A) and with (Fig. 7B) food were generated. According to the model, a plasma concentration of 200 nM, that significantly stiffens mature gametocytes following an eight-hour exposure (Fig. 7C), will be maintained for at least 8 hours in 90% and 100% of the population with a single dose of 1000 mg administered without or with food. This concentration (200 nM) is the best compromise between entire population coverage (including all modeling covariates) and the in vitro stiffening activity of TD-6450 expected to translate in a marked in vivo clearance.

**Table 1 | Scoring for the selection of the 3 confirmed hits**

| | Pre-selected hits[a] | Score[b] (safety) | Score[c] (PK) | Score (total) |
|---|---|---|---|---|
| 1 | **NITD609** | **3** | **6** | **9** |
| 2 | **(S)-(-)-Tetrahydropalmatine** | **3** | **6** | **9** |
| 3 | **TD-6450** | **3** | **3** | **6** |
| 4 | MMV-390048 | 3 | 2 | 5 |
| 5 | Halofuginone | 3 | 1 | 4 |
| 6 | Mitoquinone mesylate | 3 | 1 | 4 |
| 7 | Digitoxin | 3 | 0 | 3 |
| 8 | NVP-BGT226 | 0 | 2 | 2 |
| 9 | KDU731 | 1 | 0 | 1 |
| 10 | Chlorproguanil hydrochloride | 0 | 1 | 1 |
| 11 | Atiprimod dimaleate | 0 | 0 | 0 |
| 12 | Quisinostat | 0 | 0 | 0 |
| 13 | Abexinostat | 0 | 0 | 0 |
| 14 | Leuco methylthioninium salt | 0 | 0 | 0 |
| 15 | N-tert-butylisoquine | 0 | 0 | 0 |
| 16 | Auranofin | 0 | 0 | 0 |
| 17 | DDD498 | 0 | 0 | 0 |
| 18 | AR-42 | 0 | 0 | 0 |
| 19 | Paranyline | 0 | 0 | 0 |
| 20 | Ceritinib | 0 | 0 | 0 |
| 21 | Lanatoside A | 0 | 0 | 0 |
| 22 | Givinostat hydrochloride | 0 | 0 | 0 |
| 23 | CUDC-907 | 0 | 0 | 0 |
| 24 | MLN 576 | 0 | 0 | 0 |
| 25 | CHR-3996 | 0 | 0 | 0 |
| 26 | Cymarine | 0 | 0 | 0 |
| 27 | Proscillaridin | 0 | 0 | 0 |
| 28 | Panobinostat lactate | 0 | 0 | 0 |
| 29 | Stilbazium iodide | 0 | 0 | 0 |
| 30 | YM 161514 | 0 | 0 | 0 |
| 31 | Pirtenidine | 0 | 0 | 0 |
| 32 | BRD-7929 | 0 | 0 | 0 |

The 3 confirmed hits are in bold.

[a]44 of the 76 DRA-confirmed hits from the ReFrame library (Fig. 1) were excluded because they were not orally administered to humans.

[b]Serious adverse events reported in human subjects or no data available (Score = 0); Safe in animal models (Score = 1); No serious adverse events reported in human subjects (phase I or II) (Score = 3); No serious adverse events reported in human subjects (phase III or IV) (Score = 4).

[c]Data not available (Score = 0); Peak plasma concentration lower than 1/3 $IC_{50}$ (Score = 1); Peak plasma concentration between 1/3 $IC_{50}$ and $IC_{50}$ (Score = 2); Peak plasma concentration between $IC_{50}$ and 3 $IC_{50}$ (Score = 3); Peak plasma concentration higher than 3 $IC_{50}$ (Score = 6).

## Discussion

Using a splenic-mimetic approach adapted to high-throughput screening, we identified 82 compounds or drugs that stiffen or kill the transmission stages of *P. falciparum*, the species responsible for most severe and fatal attacks of malaria. Three of the identified active drugs are safe for use in humans and administered orally. The combination of two of them, NITD609 and TD-6450, induces high *P. falciparum* gametocyte retention rates in spleen-mimetic filters. Splenic retention results in erythrocyte clearance from the blood[10,12,46]. Drugs providing stiffening activities can therefore block malaria transmission by making transmissible forms unavailable to the blood-feeding *Anopheles* mosquito vector[9,10]. The gametocyte stage is a narrow bottleneck in the parasite cycle towards transmission to mosquitoes[8]. Blocking that transmission is already a major component of the malaria eradication effort[7] and it may take on even more importance to address the recent emergence of insecticide-resistant mosquitoes[47],

artemisinin-resistant parasites[6] and the recent upturn in global malaria incidence[1]. Our successful approach enriches the list of anti-gametocyte compounds. We identified compounds, targets, and—potentially—an anti-gametocyte drug that would not have been captured with an approach based on killing. Both the excellent safety profiles of NITD609 and TD-6450 and the physiological relevance of their gametocyte clearance effect urge for their rapid evaluation in humans in this indication. Because spleen-mimetic erythrocyte filtration has been validated against human spleens perfused ex vivo[48], NITD609 and TD-6450 are expected to actually reduce gametocyte concentrations in the circulation of human subjects.

Previously, the mechanical properties of mature gametocytes had only been explored on a small scale[13,16] and adapting the filtration of erythrocytes through layers of microspheres ("microsphiltration")[25] to high-throughput screening for gametocyte stiffeners has been a technical challenge. The effort is merited though, as microsphiltration can simultaneously analyze hundreds[28] to thousands of samples[27]. The streamlined combination of large-scale gametocyte production, microsphiltration in 384-well plates and high-content imaging eventually led to a robust assay: the screening campaign captured drugs with known gametocyte killing effects, such as methylene blue, Pf-ATPase 4 inhibitors, and Pf-PI4K inhibitors, thereby confirming the accuracy of the hit selection process. Previous screening campaigns had indeed identified methylene blue[31,33,34] and inhibitors of ATPase 4 (NITD609 and PA92), EF2 (DDD498[49]) and PI4K, (MMV390048)[50] as malaria transmission-blocking candidates. Several groups had already screened the Pathogen Box library[29–31] and our work confirms the effects of three drugs, namely MMV020081, MMV667494 and MMV030734. The two latter specifically stiffened female gametocytes in our study, confirming that a gender-biased effect of anti-gametocyte compounds is frequent[18]. We identified a cluster of 9 histone deacetylase inhibitors (HDACs) which are under clinical development for cancer, and promising candidates for other diseases, including symptomatic malaria[51], providing further evidence that the HDAC pathway is involved in malaria transmission[52]. However, we did not explore this family of compounds further because after a thorough analysis of their safety, we found it acceptable to treat cancer or potentially severe symptomatic malaria, but probably not for malaria transmission-blocking interventions[53,54]. Future optimizations may identify HDACs inhibitors more selective for *P. falciparum* and with a better safety profile[55]. Importantly, we also found several potential drug mechanisms or molecular targets. Among the hits, there were nine compounds with stiffening activities solely, specifically seven human cardiac glycosides that target the ATPase pump, and oligomycins A and B, antibiotics affecting oxidative phosphorylation through the inhibition of ATP synthase. Our findings will likely reveal new molecular pathways involved in gametocyte biomechanics. Not least, gramicidin, a pore-forming antibiotic that creates leaks in bacterial membranes, was our most powerful gametocyte killer, with an $IC_{50}$ in the low nanomolar range.

We focused post-screening explorations on the transmission-blocking potential of two drugs: NITD609 and TD-6450. NITD609 (also known as cipargamin) is an ATPase 4 inhibitor with activity against all stages of *P. falciparum*. Another ATPase 4 inhibitor, PA92, was captured by the screening campaign, confirming that these inhibitors are promising targets for transmission-blocking strategies[54]. TD-6450, a nonstructural protein 5 A (NS5A) inhibitor, has been tested in phase II clinical trials for the treatment of hepatitis C. Its antimalarial effect via mature gametocyte stiffening was revealed by our screening campaign. That TD-6450 inhibits parasite growth at micromolar range concentrations suggests that its action is predominantly parasite-specific rather than based on a putative effect on uninfected RBC. Uninfected RBC were indeed not (or only mildly) affected by exposure to the TD6450 or NITD609 in vitro (Fig. S5). Nonstructural protein 5A inhibitors in general, and TD-6450 in particular, are administered

**Table 2 | Safety and pharmacokinetics of TD-6450 in phase I clinical trials**

| Dose (mg) | N | At least one AE N (%) | At least one SAE N (%) | Death N (%) | Discontinuation due to an AE N (%) | Cmax (nM) Mean ± SD | | t½ Mean ± SD | |
|---|---|---|---|---|---|---|---|---|---|
| SAD | | | | | | | | | |
| 0.5 | 6 | 3 (50.0) | 0 (0.0) | 0 (0.0) | 0 (0.0) | 0.5 ± 0.1 | | NA | |
| 1.5 | 6 | 3 (50.0) | 0 (0.0) | 0 (0.0) | 0 (0.0) | 1.5 ± 0.4 | | 84.3 | |
| 5 | 6 | 0 (0.0) | 0 (0.0) | 0 (0.0) | 0 (0.0) | 5.37 ± 1.8 | | 71.7 ± 5.2 | |
| 15 | 6 | 1 (16.7) | 0 (0.0) | 0 (0.0) | 0 (0.0) | 21 ± 7.9 | | 77 ± 14.2 | |
| 30 | 6 | 1 (16.7) | 0 (0.0) | 0 (0.0) | 0 (0.0) | 37.4 ± 19 | | 66.9 ± 9 | |
| 60 | 6 | 0 (0.0) | 0 (0.0) | 0 (0.0) | 0 (0.0) | 123 ± 60.3 | | 61.4 ± 11.8 | |
| 120 | 6 | 1 (16.7) | 0 (0.0) | 0 (0.0) | 0 (0.0) | 173.6 ± 93.9 | | 78.4 ± 13 | |
| 240 | 6 | 0 (0.0) | 0 (0.0) | 0 (0.0) | 0 (0.0) | 278 ± 133.5 | | 64.7 ± 7.7 | |
| 500 | 6 | 2 (33.3) | 0 (0.0) | 0 (0.0) | 0 (0.0) | 243.9 ± 114.4 | | 54.9 ± 5.5 | |
| Placebo | 20 | 7 (35.0) | 0 (0.0) | 0 (0.0) | 0 (0.0) | NA | | NA | |
| MAD | | | | | | | | | |
| | | | | | | D1 | D14 | D1 | D14 |
| 60 | 8 | 4 (50.0) | 0 (0.0) | 0 (0.0) | 0 (0.0) | 97.4 ± 339 | 398 ± 146 | NA | 62.3 ± 10 |
| 120 | 8 | 3 (37.5) | 0 (0.0) | 0 (0.0) | 0 (0.0) | 215.8 ± 58 | 1014 ± 349 | NA | 66.4 ± 15 |
| 240 | 8 | 3 (37.5) | 0 (0.0) | 0 (0.0) | 0 (0.0) | 458.7 ± 203 | 1767 ± 218 | NA | 59.8 ± 12 |
| Placebo | 6 | 3 (50.0) | 0 (0.0) | 0 (0.0) | 0 (0.0) | NA | NA | NA | NA |

Abbreviations: *SAD* single ascending dose, *MAD* multiple ascending dose, *AE* adverse event, *SAE* serious adverse event, *NA* not applicable, *D1* day 1, *D14* day 14.

orally and safe in human subjects (Table S3)[55–57]. Ledipasvir, a drug displaying high homology with TD-6450, can be safely administered to pregnant women[58]. Of the 14 NS5A inhibitors included in the ReFrame library, only TD-6450 met the hit definition. TD-6450 is the only heterodimeric NS5A inhibitor which may explain its enhanced activity. Nevertheless, a class effect cannot be excluded at this stage. The maximal gametocyte retention rates observed with TD-6450 were lower than those of NITD609, reminiscent of a partial antagonism. Gametocyte populations contain males and females and a gender-biased effect has been described with anti-gametocyte drugs[59]. The readout-dependent differences in efficacy observed with TD-6450 in our study may explain the apparently partial activity, which may thus be complete on males and weak or absent on females. A robust confirmation of this point is required.

Despite the relatively high $IC_{50}$ observed in vitro, a therapeutic window for TD-6450 likely exists. The modeling data predict that almost 90% of a population will be exposed to drug plasma concentrations expected to reduce the density of circulating gametocytes. The three main determinants of malaria transmission are the concentration of gametocytes in peripheral blood (gametocytemia), their potential sequestration in human skin, and the sexual commitment during the multiplication cycle. Of those, only gametocytemia robustly correlates with transmission[60–63]. In a recent study[60], no or very few *Anopheles* were infected by blood containing 30 or less gametocytes/μl, and the number of infected *Anopheles* was at least 10 times lower at 50 gametocytes/μl than at 200–250 gametocytes/μl. Therefore, a drug or a drug combination inducing the clearance of 70–80% of gametocytes may impact transmission, regardless of its effect on parasite viability; even alive, a stiffened gametocyte retained in the spleen is indeed inaccessible to the mosquito bite. The long half-life of TD-6450 and the persistence of its activity after removal of drug pressure are expected to facilitate a sustained transmission-blocking effect.

In the absence of a validated animal model for the splenic retention of mature *P. falciparum* gametocytes[48], a study in humans is the next step. In a recent human challenge study, a sustained circulation of mature gametocytes was observed following the treatment of volunteers with piperaquine. That result lays a pathway to a fast and powerful evaluation of the transmission-blocking potential of old or new drugs[64]. For drugs like those identified in here, potentially able to block transmission by inducing splenic gametocyte clearance, the readout for such an evaluation would be the simple measure of gametocytemia following drug administration. Preclinical information very important for the development of anti-gametocytic drugs such as Standard Membrane Feeding Assays and studies in mice are dispensable prerequisites in the specific context of our approach. A decrease in gametocytemia is indeed directly related to reduced transmission, and we provide robust results suggesting that TD-6450 and NITD609 may reduce gametocytemia. Developing drugs to block the transmission of malaria is confronted to ethical, pharmaceutical, and logistical challenges. If the potential transmission-blocking effect of NITD609 and TD-6450 is confirmed in a clinical trial, adding these safe drugs to the armamentarium may contribute to the momentum of an original approach aiming at a sustained control of malaria.

## Methods
### Libraries and screening conditions
The Pathogen Box library (400 compounds, provided by MMV-Medicine for Malaria Ventures, Geneva, Switzerland), the Kinase Inhibitors Box library (350 compounds, provided by GSK-Glaxo SmithKline DDW Unit, Tres Cantos, Spain), and the ReFrame library (12,805 compounds provided by the California Institute for Biomedical Research-Calibr, La Jolla, CA, USA) were used according to the providers' instructions. DMSO-solubilized compounds (10 mM stock concentration) were loaded in 384-well plates (Seahorse, cat. no. 201035–100). Two columns were left empty for the controls. For each compound, 5 nL were pre-spotted in a single well to reach a final concentration of 1.11 μM. The final volume for each well was 45 μl. Controls were prepared as follows: 22.5 nL of DMSO were loaded in 16 wells as negative controls (0.05% final concentration); 22.5 nL of NITD609 (provided by MMV, Geneva, Switzerland) 1 mM were loaded in 8 wells (0.5 μM final concentration), 17 nL of Calyculin A (Sigma, cat. no. C5552) 0.2 μM were loaded in 6 wells (75 nM final concentration) and 225 nL of Gentian Violet (MolPort, cat. no. 002-133-551) 10 mM were loaded in the remaining 2 wells (50 μM final concentration). Plates were prepared in triplicate for the Pathogen Box and sextuplicate for the Kinase Inhibitors Box.

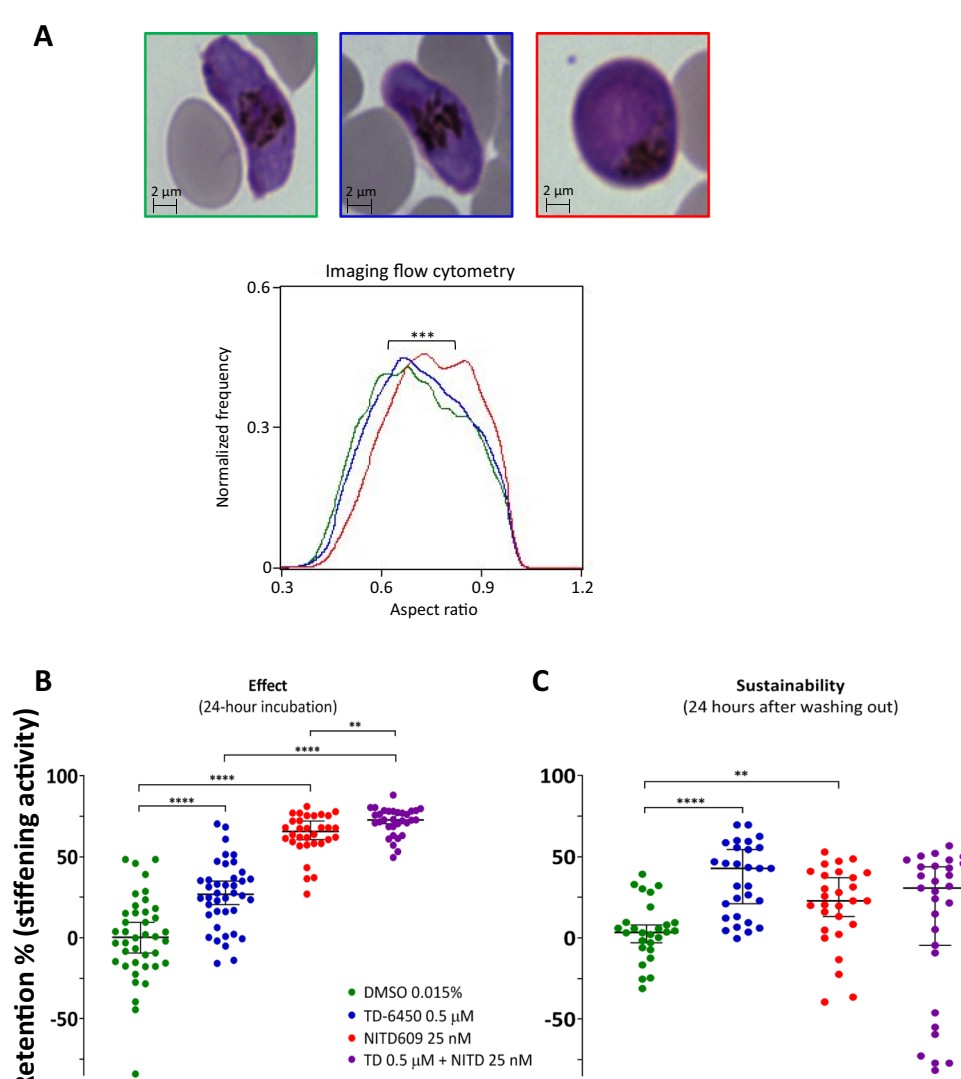

**Fig. 4 | Activity of NITD609 and TD-6450 upon post-screening analysis.**
**A** Giemsa-stained erythrocytes infected by a mature gametocyte of *P. falciparum* exposed for 24 hours to DMSO (negative control, green border), TD-6450 5 μM (blue border) and NITD609 1 μM (red border). Circularity of gametocytes by imaging flow cytometry was compared measuring the aspect ratio showing a significant difference between DMSO and NITD609. The experiment was performed once. **B** Cumulative dot-plot of 5 microsphiltration experiments where stage V gametocytes of *P. falciparum* (3D7 pULG8-GFP strain) were exposed for 24 hours to TD-6450 (blue), NITD609 (red) or the combination of both (purple). **C** Cumulative dot-plot of 4 microsphiltration experiments where gametocytes were exposed to the drugs for 24 h and kept in culture for an additional 24 hours after removing the drug. Dots indicate the retention rate of single wells of 96-well microsphiltration

plates. For **B**, **C**, each experiment was performed with a single gametocyte induction and a single 8-well column for each condition was loaded, unless the cultured gametocyte population was not large enough to fill the entire column. Numbers of repetitions are, from left to right: 40 (8 for each experiment), 40 (8 for each experiment), 32 (8 for each of 4 experiments), 32 (8 for each of 4 experiments). Boxplots indicate medians and IQRs. Exposure to DMSO (green) was the negative control. $P$ and $F$-values after the one-way ANOVA test were lower than 0.0001 and equal to 110 (B) and equal to 0.0005 and 6.365 (**C**), respectively. Individual $p$-values legend: *$p = 0.05–0.01$, **$p = 0.01–0.001$, ***$p = 0.001–0.0001$, ****$p < 0.0001$. Individual $p$-values, when higher than 0.0001, are: panel B, NITD609 vs combination, 0.0058. Panel C, DMSO vs NITD609, 0.0096. Source data are provided as a Source Data file.

## Culture of *Plasmodium falciparum* asexual and sexual blood stages

*Plasmodium falciparum* NF54 strain parasites (used for screening campaign) were cultured at 4% hematocrit in RBCs from informed blood donors (blood group A) as previously described[65]. Parasites were taken from the -80 °C freezer, then rapidly melt for 1 minute in a 37 °C water bath. A saline solution was gently added to avoid the lysis. Firstly, NaCl dissolved at 12% in distilled water (1/5 of the volume measured in the cryovial) and then NaCl dissolved at 1.6% (9 times the cryovial's volume). After that, fresh RBCs were added to allow parasites regrowth. Infected RBCs were kept in RPMI medium (Sigma-Aldrich, cat. no. R4130), supplemented with 10% of decomplemented human serum, hypoxanthine 50 mg/l (Sigma-Aldrich, cat. no. H9636). Biobank

of Castilla y Leon and Centro de Transfusiones de Madrid as providers of human red blood cell concentrates. The production of gametocytes was induced from a 0.5% ring-stage parasitemia (day 0). Medium was changed daily to induce gametocytes differentiation, as previously described[66]. At day 17, gametocytes were concentrated using a density gradient medium (NycoPrep 1.077, Progen cat. no. 1114550)[27]. Parasitemia was adjusted by FACS (SYBR-Green and MitoTracker double staining, 3–5% targeted). Hematocrit was adjusted to 0.5% by adding fresh RBCs and controlled by manual counting in a Neubauer chamber (Celeromics, cat. no. 717810). Finally, the gametocyte suspension was poured (45 μL/well) into the compound-containing 384-well microplate, then incubated 24 hours both for the screening campaign and the dose-response analysis. Both asexual and sexual stages were

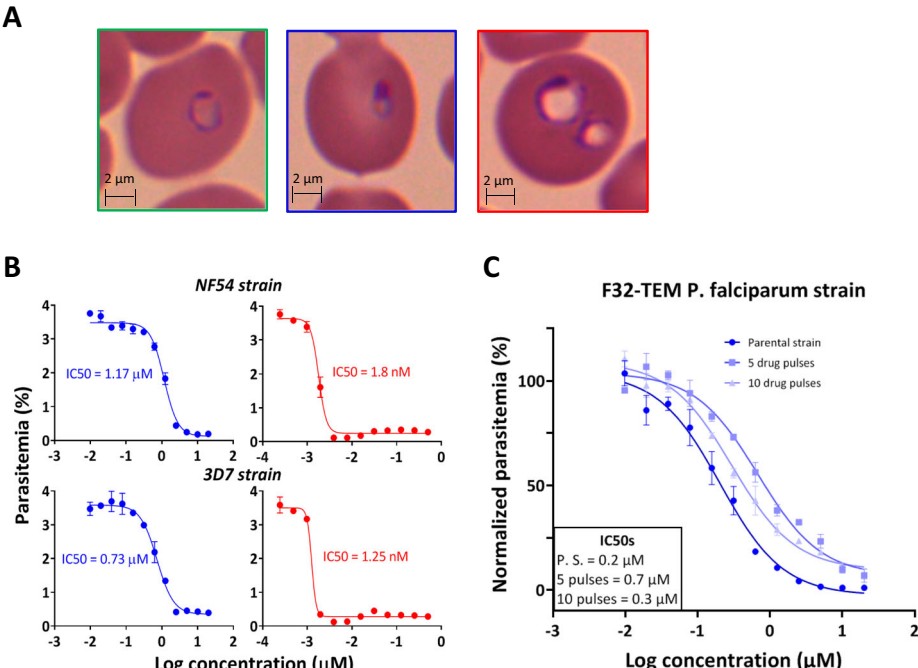

**Fig. 5 | Activity of NITD609 and TD-6450 on asexual parasites. A** Giemsa-stained erythrocytes infected by a ring-stage of *P. falciparum* exposed to DMSO (negative control, green border), TD-6450 5 μM (blue border) and NITD609 1 μM (red border). The optical microscopic observations were repeated 5 times. **B** Dose-response of synchronized ring-stage *P. falciparum* (NF54 and 3D7 pULG8-GFP strains), exposed for 48 h to TD-6450 (blue) and NITD609 (red). **C** Dose-response of F32-TEM strain either before (dark blue), or after 5 (blue) or 10 (light blue) drug pulses with TD-6450. Each point shows the mean (±SD) of 3 replicates. Source data are provided as a Source Data file.

cultivated in RPMI 1640 supplemented with 25 mM HEPES, 50 mg/L hypoxanthine, 50 μg/mL gentami-cin, 10% pooled heat-inactivated human serum (A⁺). Each batch of serum was obtained by pooling 10 serum bags from different donors. Serum was kept at −20 °C and thawed in a warmed water bath before use. A new serum batch was prepared when the previous was completed. For dose-response analysis and microfluidics, both the 3D7-pULG8-GFP[67] and the NF54 strain (BEI Resources, cat. no. MRA-1000) were tested. The 3D7 strain was cultivated as described above with the addition of 0.2% NaHCO3 and 2.5 nM of WR99210 for both asexual and sexual stages.

**Large scale microsphiltration**

Large scale microsphiltration was performed as previously described[27]. Briefly, microsphiltration plates were prepared using a Biomek NX workstation (Beckman, cat. no. 989402) to load 15 μl of 25–45μm microspheres then 10 μl of a mixture of 30% 5–15μm and 70% (w/w) 15–25μm microspheres. The 384-well filter plates were then stored at -20 °C until use. In some of the 384-well microsphiltration plates leakage of microspheres occurred in one or more wells. The wells with leakages were sealed. Plates displaying more than 5 sealed wells were not used. Imaging plates were prepared by loading 30 µL of staining solution in each well of a collagen-coated plate. A different staining solution was used for each readout (1 and 2). Readout 1 counted both male and female gametocytes, using nucleic acid (Syto40, Thermo Fisher cat. no. S11351) and viability (MitoTracker Deep Red, Thermo Fisher cat. no. M22426) staining, and total RBCs using a cell membrane stain (CellMask Orange, Thermo Fisher cat. no. C10045). Readout 2 selectively quantified the retention and killing of female gametocytes. A specific female gamete marker (anti-pfs25, clone 4B7, BEI Resources cat. no. MRA-315), anteriorly coupled with Cy3 fluorescent dye (GE Healthcare, cat. no. PA33000), was dissolved in ookinete medium (RPMI medium with 25 mM HEPES, 50 mg/liter hypoxanthine, 2 g/liter sodium bicarbonate, 100 μM xanthurenic acid, 20% human

serum) to induce female gamete activation[18]. A different cell membrane stain (CallMask Deep Red, Thermo Fisher cat. no. C10046) was also added to count all RBCs. An automated microsphiltration protocol was performed using Biomek NX Workstation (Beckman, cat. no. 989402) connected to a vacuum pump. Automated microsphiltration was performed as previously described[27]. Imaging plates were incubated overnight at room temperature, then read in an OperaPhenix high-content confocal imaging system (Perkin Elmer, Waltham, MA, USA) and analyzed with Harmony high-content analysis software v. 5.0.

**Statistical analysis and *Z′* quantification**

From the analysis of imaging plates, the percentage of gametocyte infection was calculated based on the number of gametocytes divided by the number of total cells. This was done for upstream (UP) and downstream (DS) samples. Killing was calculated using the following formula:

$$\text{Killing rate} = 100 - ((\%UP * 100)/\text{Average } \% \text{ UP negative controls}) \tag{1}$$

Retention rate was calculated as previously described[25] with the following formula:

$$\text{Retention rate} = 100 * ((\%UP - \%DS)/\%UP) \tag{2}$$

Killing (*x* axis) and retention (*y* axis) rates were then graphed. A regression line with a 95% prediction band was calculated using GraphPad Prism 5.1 software. Outliers were excluded from linear regression calculation. Only those compounds that fell outside the upper prediction band in both readouts were finally selected as hits. Analyses were done plate by plate. For each single screening plate, the *Z′* parameter was calculated as previously described[68] with the

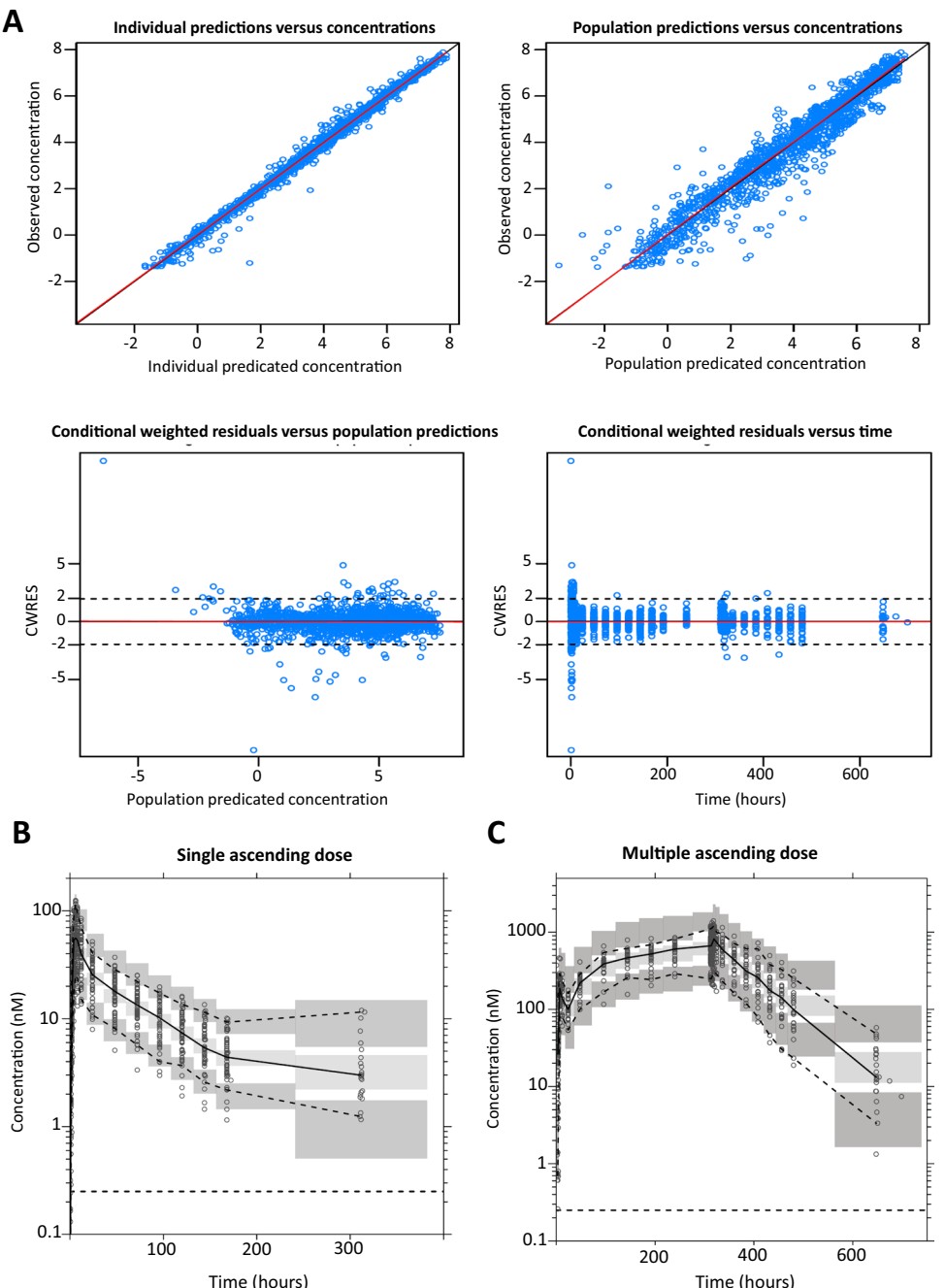

**Fig. 6 | TD-6450 modeling based on phase I results.** Goodness-of-fit (**A**) of the final pharmacokinetic model. Simulation-based (*n* = 2000 independent simulations) diagnostics (pvcVPC), displaying the predictive performance of the final PK model, stratified on single (**B**) vs multiple (**C**) dosing. PK data used to realize the modeling were made available by Theravance Biopharma. Source data are provided as a Source Data file.

following formula:

$$Z' = (1 - ((3 * SD_{PC}) + (3 * SD_{NC})) / |RR_{PC} - RR_{NC}|) \qquad (3)$$

where RR is the retention rate average, SD the standard deviation, PC the positive control (NITD609 0.5 µM) and NC the negative control (DMSO 0.05%).

Regarding the statistical analysis of post-screening experiments, an ordinary one-way ANOVA test was preliminary performed to validate it, when two-tailed *P*-value was inferior to 0.001. The ANOVA was followed by a paired t test to calculate the two-tailed *P*-values of individual differences (showed with asterisks in the figures).

### Dose-response analysis and IC$_{50}$ determination

Dose-response analyses (DRA) were done for killing effect and stiffening activity for each selected hit as defined above. Results from OperaPhenix were graphed with the x axis representing the compound concentration and the *y* axis the killing or retention rate. Using GraphPad 5.1 Prism software (GraphPad Software, San Diego, CA, USA), the results were transformed into logarithmic form and a log(agonist) vs response curve was tracked (variable slope). The concentration at which 50% of killing or retention is inhibited (IC$_{50}$) was calculated by

**Table 3 | Final pharmacokinetic parameter estimates of TD-6450**

| Pharmacokinetic parameters | [a]Population estimates ([b]%RSE) | [b]SIR median (95%CI) |
|---|---|---|
| Fixed effects | | |
| F (%) | 1 *fixed* | |
| MTT (h) | 2.30 (4.14) | 2.30 (2.13 to 2.50) |
| CL/F (L/h) | 12.1 (6.19) | 12.1 (10.8 to 13.6) |
| Vc/F (L) | 537 (7.89) | 539 (464 to 626) |
| Q/F (L/h) | 28.4 (11.4) | 28.5 (22.5 to 35.5) |
| Vp/F (L) | 444 (7.36) | 445 (384 to 513) |
| Covariate relationships | | |
| Dose on F | -0.247 (7.92) | -0.241 (-0.207 to -0.284) |
| Dose on CL/F | 1.35 (17.9) | 1.35 (0.873 to 1.80) |
| Time dep. F | 0.439 (12.5) | 0.441 (0.355 to 0.569) |
| Food on F | 0.685 (15.0) | 0.683 (0.475 to 0.885) |
| Food on MTT | 0.933 (10.3) | 0.927 (0.696 to 1.19) |
| Random effects | | |
| IIV F | 0.150 (16.7) | 0.151 (0.110 to 0.207) |
| IIV MTT | 0.0683 (10.3) | 0.0683 (0.0551 to 0.0830) |
| IIV CL/F | 0.0148 (21.5) | 0.0147 (0.0103 to 0.0223) |
| IIV Vc/F | 0.0391 (17.3) | 0.0394 (0.0246 to 0.0514) |
| IIV Vp/F | 0.0713 (12.9) | 0.0719 (0.0568 to 0.0919) |
| IOV F | 0.0631 (13.9) | 0.0622 (0.0445 to 0.0794) |
| IOV MTT | 0.0866 (17.1) | 0.0870 (0.0613 to 0.118) |
| RUV | 0.242 (4.48) | 0.0568 (0.0520 to 0.0624) |

[a]Computed population mean parameter estimates from NONMEM were calculated for a typical individual with a body weight of 80 kg, at steady-state when receiving a high oral dose (fully induced clearance) in a fasting state.
[b]Computed from sampling importance resampling (SIR; 2000 samples, 1000 resamples) and presented as median estimates (2.5th to 97.5th percentiles).
Abbreviations: *SIR* sampling importance resampling, *RSE* relative standard deviation, *F* relative bioavailability, *MTT* mean absorption transit time over 4 transit compartments, *CL/F* oral elimination clearance, *Vc/F* apparent central volume of distribution, *Q/F* inter-compartment clearance, *Vp/F* apparent peripheral volume of distribution, *IIV* inter-individual variability presented as variance estimates, *IOV* inter-occasion variability presented as variance estimates, *RUV* residual error presented as variance estimates.

the software from the equation of the curve:

$$y = B + (T - B)/(1 + 10^{(\text{LogIC50}-x)} * \gamma) \qquad (4)$$

where B is bottom value, T is top value and ɣ is the hillslope.

**Post-screening validation with 96-well plate microsphiltration**
Both fixed-dose and dose-response analyses were performed according to a second protocol (Paris laboratory) to validate the selected hits from primary screening. Briefly, the gametocyte suspension was loaded into a 96-well empty plate (200 μL/well). For each dose-response, first, well volume was doubled and compound was added to obtain 10 μM final concentration. For the dose-response analysis, the gametocyte suspension was then manually serially diluted 1/2 for 12 points, obtaining 5 nM as the lower concentration. The same operation was done with DMSO to have a negative control. After 24 hours of incubation at 37 °C, gametocyte suspensions were loaded into a 96-well filtration plate[28] and microsphiltration was manually performed at 37 °C. Up and downstream samples were incubated for 30 min with

Hoechst 33342 nuclear staining (1/1000 diluted in PBS, Thermo Fischer cat. no. H3570), washed, resuspended in 200 μL of PBS and then loaded into an empty 96-well plate before FACSCanto (BD Biosciences, Franklin Lakes, New Jersey, USA) quantification. Data from cytofluorimetry were analyzed with FlowJo V12. As per previously described experiences with this technique, to avoid the risk of microbead layers saturation, a microsphiltration experiment was repeated when the upstream gametocytemia exceeded 10%[25,69].

**Imaging flow cytometry analysis**
Imaging flow cytometry using ImageStream X Mark II (Amnis part of Luminex) was performed to determine the aspect ratio of erythrocytes infected by a mature gametocyte of *P. falciparum* by using brightfield images (60x magnification) processed with computer software (IDEAS v 6.2, AMNIS). The aspect ratio is the ratio of the minor axis divided by the major axis and describes how round or oblong an object is. Focused cells and single cells were selected to analyze the aspect ratio feature to document the shape of the cells.

**TD-6450 dose-response analysis on asexual parasites**
*Plasmodium falciparum* 3D7-pULG8-GFP or NF54 strains were synchronized by sorbitol lysis[70]. Rings were then diluted to 0.5% parasitemia and loaded into an empty 96-well plate (200 μL/well). First, well volume was doubled and compound was added to obtain 20 μM (TD-6450) and 500 nM (NITD609) final concentrations. The ring suspension was then manually serially diluted 1/2 for 12 points, obtaining 9.77 nM (TD-6450) and 0.24 nM (NITD609) as lowest concentrations. The same operation was done with DMSO to have a negative control. After 48 hours of incubation, the plates were centrifuged and RBC pellets were incubated for 30 min with Sybr-G (1/1000 diluted in PBS, Thermo Fisher cat. no. S7567). After PBS washing (2x), pellets were resuspended in 200 μL of PBS and parasitemia was quantified by FACSCanto and analyzed with FlowJo V12. IC50s were determined using GraphPad Prism 5.1 as mentioned. Images from optical microscopy were taken using a Leica inverted microscope and analyzed with Las X software.

**TD-6450 phase I clinical study**
This was a phase I, double-blinded, randomized, placebo-controlled, Single Ascending Dose (SAD) and Multiple Ascending Dose (MAD) study to evaluate the safety and tolerability of TD-6450 in healthy subjects (primary objective) and the pharmacokinetics (secondary objective). The study is available in the website clinicaltrial.gov (identifier NCT02022306), it started on February the 4th 2014 (first subject enrolled) and was concluded on August the 18th 2014 (last subject concluded), in the Healthcare Discoveries (a subsidiary of ICON Development Solutions) in San Antonio, Texas, USA. Healthy, non-smoking subjects with no history of significant medical conditions, with a BMI included between 18 and 30 kg/m², who weight at least 50 kg, between 18 and 60 years old, were selected for this study. Subjects were able and willing to give an informed consent. No bias was reported in the selection process. The protocol and all amendments for this study and all accompanying material that was provided to subjects (including advertisements, subject's information sheets, and descriptions of the study used to obtain informed consent) were submitted by the investigator to the centralized Institutional Review Board (IRB), namely the IntegReview, established in 1999 at Austin (Texas, U.S.A.) and acquired by Advarra in 2020. Documented approval was obtained before the study was started, on March the 28th 2014. This study was conducted in accordance with the protocol, the principles of the International Conference on Harmonization of Technical Requirements for Registration of Pharmaceuticals for Human Use (ICH) Guideline for Good Clinical Practice (GCP), the United States Code of Federal Regulations, the principles of the World Medical Association Declaration of Helsinki, Ethical Principles for

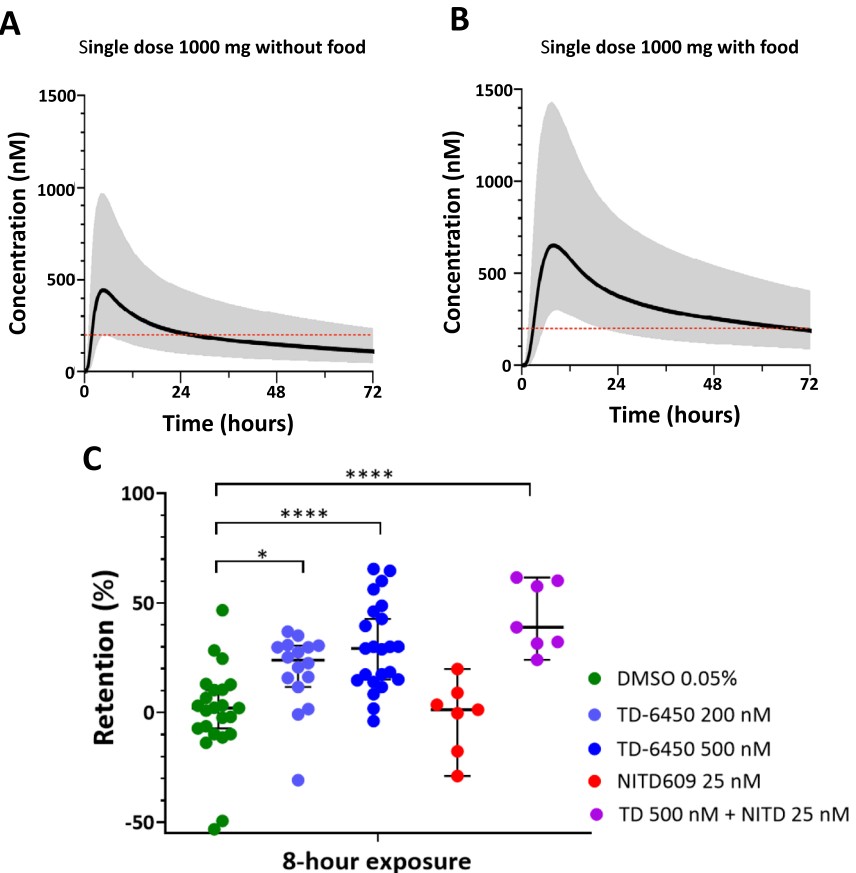

**Fig. 7 | TD-6450 showed an activity at concentrations that can be reached in healthy subjects receiving a single dose.** Simulated population pharmacokinetic profiles of a single oral dose of TD-6450 when administered in a fasting state (**A**) and with food (**B**), in healthy volunteers. The black solid lines are the population mean profiles and the shaded gray area show the 90% prediction interval. The red dashed lines indicate a concentration of 200 nM, associated with significant stiffening of mature gametocytes. PK data used to realize the modeling were made available by Theravance Biopharma. **C** Cumulative dot-plot of 3 microsphiltration experiments where gametocytes were exposed to the drugs for 8 h, diluted by a factor of 10 and then filtrated after 24 h. Dots indicate the retention rate of single wells of 96-well microsphiltration plates. Each experiment was performed with a single gametocyte induction and a single 8-well column for each condition was loaded, unless the cultured gametocyte population was not large enough to fill the entire column. Numbers of repetitions are, from left to right: 23 (8 for the first 2 experiments, 7 for the 3rd), 16 (8 for each of the first 2 experiments), 23 (8 for the first 2 experiments, 7 for the 3rd), 7 (3rd experiment only) and 7 (3rd experiment only). Boxplots indicate medians and IQRs. Exposure to DMSO (green) was the negative control. *P* and *F*-values after the one-way ANOVA test were lower than 0.0001 and equal to 12.09, respectively. Individual *p*-values legend: *$p = 0.05$–0.01, **$p = 0.01$–0.001, ***$p = 0.001$–0.0001, ****$p < 0.0001$. Individual *p*-values, when higher than 0.0001, are: DMSO vs TD-6450 200 nM, 0.0284. Source data are provided as a Source Data file.

Medical Research Involving Human Subjects, and all applicable regulatory requirement.

Primary and secondary outcomes were pre-defined as follows: Safety and tolerability as a primary outcome, pharmacokinetics as a secondary outcome. Safety and tolerability were evaluated using standard measures, including adverse event (AE) monitoring, clinical laboratory tests (hematology, serum chemistry, coagulation, and urinalysis), vital signs, physical examinations, and 12-lead electrocardiograms (ECGs). Adverse events were assessed by the investigator and defined as any untoward medical occurrence in a patient or clinical investigation subject administered a pharmaceutical product and which did not necessarily have a causal relationship with this treatment. Pharmacokinetics were evaluated by the collection of blood samples from each subject at following time points: predose (within 30 min before dosing, only in day 1 60 min before dosing), and 0.5,1,2,3,4,6,8 and 12 hours after dosing. Blood samples were analyzed to calculate the following PK parameters: $AUC_{0-24}$: area under the plasma concentration versus time curve, from time zero to 24 hours post infusion, $AUC_{0-t}$: area under the plasma concentration versus time curve, from time zero to the last sample with quantifiable analyte concentration, $AUC_{0-\infty}$: area under the plasma concentration versus time curve, from time zero to infinity, $C_{max}$: peak plasma concentration, $T_{max}$: time to reach $C_{max}$, terminal elimination half-life ($t_{1/2}$), CL/F: oral plasma clearance, Vz/F: apparent volume of distribution during the terminal phase, Ae (amount excreted in urine): the cumulative amount excreted in urine from time zero to the last sample with measurable analyte concentration, Fe: fraction of dose excreted in urine (TD-6450 only) (Day 1 and 14 MAD only), CLr: renal clearance.

This study was conducted in two parts, Part A (SAD) and Part B (MAD). For the Part A, the age of enrolled subjects ranged from 20 to 60 years, with a mean age range of 32.1 to 46.2 years. Most subjects were male (73 total subjects) and most were white (55 subjects) or black (22 subjects). For Part B of the study, ages ranged from 23 to 60 years with a similar mean age across treatment groups. Most subjects were male (25 total subjects) and all were either white (18 subjects) or black (12 subjects). For each part of the study, the mean body mass index (BMI) and creatinine clearance were similar across treatment groups. No subject had a clinically significant medical history of relevance. Part A: Healthy subjects were sequentially enrolled and randomly assigned to each dose cohort (up to 10 dose-ascending cohorts were planned, including a fed arm in the food effect cohort) to receive either a single dose of TD-6450 or placebo. For each dose cohort, 8 subjects were randomized in a 3:1 ratio (6 subjects received TD-6450 and 2 subjects received placebo). The starting dose of

TD-6450 was 0.5 mg with a planned maximum dose of 1000 mg. Upon review of the 500-mg cohort PK data, the decision was made to not escalate to the 1000-mg cohort because of exposure saturation at ≥ 500 mg. The following doses were administered: 0.5 mg, 1.5 mg, 5 mg, 15 mg, 30 mg, 60 mg, 120 mg, 240 mg, and 500 mg. Part B: Healthy subjects were sequentially enrolled in 3 ascending dose cohorts (TD-6450 60 mg, 120 mg, and 240 mg) and randomly assigned to receive either TD-6450 or placebo as once daily doses for 14 days. For each dose cohort, 10 subjects were randomized in a 4:1 ratio (8 subjects received TD-6450 and 2 subjects received placebo). Blinded safety, tolerability, and available PK data after each dose cohort were reviewed by the Safety Data Review Committee (SDRC) before escalation to the next dosing cohort for both part A and B. Subjects of the clinical study were randomly assigned to each cohort. Randomizations codes were assigned to the subjects. All study subjects, study investigators, and the sponsor's staff involved in the conduct of the study were blinded to treatment assignment. The only personnel who had access to the randomization code before database lock were: The pharmacist or pharmacy staff member who prepared drug or placebo capsules for administration to the subject according to the randomization code and the bioanalytical contract laboratory that analyzed the PK samples (plasma and urine).

Study drug for each subject was appropriately labeled to facilitate the correct administration to the assigned subject. All documentation regarding subjects' treatment assignment (i.e. randomization codes, drug accountability records) were maintained in a secure area so that only the unblinded pharmacy staff had access to them, and these individuals were not involved in any aspect of subject management, case report form (CRF) entry, monitoring, or reporting required by the protocol, other than for dispensing records. In the event of a medical emergency or unblinding for dose escalation purposes, the identity of the study drug would have been unblinded at the clinical site. If a medical emergency arose, the randomization code for an individual subject would have broken using a set of sealed emergency codes provided to the pharmacist or the authorized staff member.

Due to the exploratory nature of this study, no formal power or sample size calculations were used to determine cohort size. A sample size of 88 healthy subjects (8 subjects per cohort) for the SAD study and a sample size of 30 healthy subjects (10 per cohort) for the MAD study should provide adequate characterization of PK and safety assessments within this setting. At screening, eligible subjects were 18 to 60 years of age (inclusive), had a body mass index 18–30 kg/m2 (inclusive), weighed at least 50 kg, and were in good health as judged by the absence of clinically significant diseases or clinically significant abnormal laboratory values (Screening or Day −1). The study was conducted in the Healthcare Discoveries (a subsidiary of ICON Development Solutions) in San Antonio, Texas, USA.

## PK modeling
Data were pooled from two healthy volunteer studies, one single ascending dose (SAD) and the other multiple ascending dose (MAD). Observed concentration data were transformed into their natural logarithms and evaluated using nonlinear mixed-effects modeling in NONMEM v7.4 (Icon Development Solution, Ellicott City, MD). Pirana v3.0, Perl-speaks-NONMEM (PsN) v4.8.1, and Xpose version 4.7.1, were used for automation, model evaluation, and diagnostics during the model building process.

Different structural distribution models were evaluated (1-, 2-, and 3-compartment models) as well as different absorption models (first-order absorption and transit-compartment ($n = 1$–10) models). Relative bioavailability was fixed to unity for the population to evaluate inter-individual variability and the influence of covariates in the same parameter. Body weight was evaluated as a fixed allometric function (exponent of 0.75 for clearance parameters and 1.0 for volume parameters), cantered on a typical patient weighing 80 kg. Other biologically plausible covariate-parameter relationships (i.e. sex and age on all parameters, dose amount and duration of treatment on elimination clearance, absorption rate and relative bioavailability, concomitant food intake on absorption rate and relative bioavailability, and creatinine clearance on elimination clearance) were evaluated with a stepwise addition ($p < 0.05$) and elimination ($p < 0.001$) approach. All significant covariates in the forward step were evaluated with both linear and nonlinear functions, and the best-performing model was retained for the backward elimination step. Pharmacokinetic parameters were assumed to be log-normally distributed, and inter-individual variability (IIV) and inter-occasion variability (IOV) were implemented as exponential functions. Estimated IIV below 10% was removed in the final model, and IOV was estimated only on absorption parameters.

Model fit was evaluated primarily by the objective function value (OFV; calculated by NONMEM as proportional to $-2 \times$ log-likelihood of the data). Model discrimination between two hierarchical models was determined by a likelihood ratio test, based on the Chi-square distribution of the OFV (i.e. $p$-value $< 0.05$ corresponding to $\Delta OFV > 3.84$, at a 1 degree of freedom difference). Goodness-of-fit was used to evaluate the descriptive performances of developed models. Prediction- and variability-corrected visual predictive checks (pvcVPC, $n = 2000$ simulations) were used to assess the predictive performance of developed models.

### Reporting summary
Further information on research design is available in the Nature Portfolio Reporting Summary linked to this article.

## Data availability
All data associated with this paper are present in the main manuscript or in the Supplementary Materials. The results from the primary in vitro screen of the ReFrame library generated in this study have been deposited in the Reframe database (https://reframedb.org/assays/A00279) under accession code A00279. The Phase I clinical study (ID: NCT02022306) described in this paper is deposited in the clinical-trials.gov database (https://clinicaltrials.gov/ct2/results?cond=&term=NCT02022306&cntry=&state=&city=&dist=) under the accession code NCT02022306. Source data are provided with this paper.

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

## Acknowledgements

We thank Medicines for Malaria Venture (MMV) for providing the Pathogen Box and GSK for providing the Kinase Inhibitors Box. We thank prof. Pietro Alano for providing the *P. falciparum* strain 3D7 pULG8-GFP and Françoise Benoit-Vical for providing the strain F32-TEM. We are very grateful to Jeremy Burrows and Didier Leroy from MMV and to Laurent Fraisse from Drugs of Neglected Diseases Initiative (DNDi) for their critical review of the entire project. We thank Theravance Biopharma Inc. as sponsor of the clinical trial described here and their former senior vice presidents Brett Haumann (sponsor signatory of the study) and Philip Worboys for collecting and analyzing the data of the trial and made them available for this paper, as well as the Executive Director Wayne Yates for his constant support. We also thank Maria Jesùs Almela-Armendariz and Jorge Fernandez Molina from GSK for their technical support. We acknowledge the Bill & Melissa Gates Foundation for the grant OPP1123683 and the Tres Cantos OpenLab Foundation for the grant TC180 obtained by P.B. We acknowledge as well the support from NIH (R01HL154150) to M.D. and the support from HRA Pharma and André Ulmann to J.D.

## Author contributions

M.C. wrote the manuscript, contributed to screening assay setup, and performed the screening and post-screening experiments. J.D. set up the screening assay. I.G.-B. provided technical support for the primary screening campaign. B.B., A.F.-M., A.S. and L.D. provided technical support for post-screening validation experiments. P.A. and C.R. supervised the imaging flow cytometry experiments. M.D. performed imaging flow cytometry experiments. M.D. established the interpretation of screening data based on regression analysis of primary screening. P.G. contributed to the primary screening analysis. C.W.M. and M.V.H. provided technical support and information for everything related to the ReFrame library. P.A.N. critically reviewed the results and the manuscript. F.J.G. and L.S.A. jointly supervised the primary screening campaign in GSK DDW. P.B. conceived the entire project, supervised and coordinated the experimental activity and wrote and edited the manuscript. All authors critically reviewed the manuscript.

## Competing interests

The authors declare no competing interests.

## Additional information

[1]Université Paris Cité, Inserm, UMR-1134, Biologie Intégré du Globule Rouge, 75015 Paris, France. [2]SYNSIGHT, 91000 Evry-Courcouronnes, France. [3]Mahidol-Oxford Tropical Medicine Research Unit, Faculty of Tropical Medicine, Mahidol University, 10400 Bangkok, Thailand. [4]Centre for Tropical Medicine and Global Health, Nuffield Department of Medicine, University of Oxford, Oxford, UK. [5]Mycology Reference Laboratory, National Centre for Microbiology, Instituto de Salud Carlos III, 28222 Madrid, Spain. [6]Global Health Medicines R&D, GlaxoSmith Kline (GSK), 28760 Tres Cantos, Spain. [7]Laboratory of cellular and molecular mechanisms of hematological disorders and therapeutic implications, INSERM, 75014 Paris, France. [8]Laboratoire d'Excellence GR-Ex, Paris, France. [9]Department of Materials Science and Engineering, Massachusetts Institute of Technology, MA 02139 Cambridge, USA. [10]Calibr, a division of The Scripps Research Institute, La Jolla, CA 92037, USA. [11]Laboratoire d'Hématologie générale, Hôpital Universitaire Necker Enfants Malades, Assistance Publique–Hôpitaux de Paris (AP-HP), 75015 Paris, France. [12]Department of Infectious & Tropical Disease, AP-HP, Necker Hospital, 75015 Paris, France. [13]Centre Médical de l'Institut Pasteur (CMIP), Institut Pasteur, 75015 Paris, France. [14]These authors contributed equally: Mario Carucci, Julien Duez. ✉e-mail: pierre.buffet@inserm.fr

