## [Peer review file · Nature Communications]

REVIEWER COMMENTS

Reviewer #1 (Remarks to the Author):

The submission describes the application of a recently developed microspherultration (microsphere filtration) process that was used to screen three chemical libraries for compounds that induce stiffening in *Plasmodium falciparum* infected erythrocytes: specifically late-stage gametocytes. In malaria, the spleen is responsible for trapping and removing infected erythrocytes from the blood. Only the early ring stages and late-stage gametocytes can sufficiently deform themselves to squeeze through the narrow interendothelial slits within the spleen. The microspherultration process described can mimic the function of the spleen in removing infected erythrocytes from blood. The process was developed and designed to rapidly identify compounds that can induce stiffening of late-stage gametocytes thus tapping and removing them from the transmission cycle.

Targeting late-stage gametocytes and screening large chemical libraries to identify potential transmission blocking (TB) drugs have been widely reported, but this is the first example of a spleen-mimetic being used to screen more than 12000 compounds from the ReFrame drug repurposing library. Presumably, any ReFrame hits could be further developed since PK and safety in humans is well characterized. The results of the screening identified three compounds that display a retention (stiffening) effect: the antimalarial NITD609; an antiviral NS5A inhibitor, TD-6450; and L-THP, an alkaloid with anxiolytic and sedative effects.

The microspherultration approach is inventive and a technical achievement that has been previously described in detail (Methods Mol. Biol. 2013; 923:291 and Nat. Protoc. 2018 13:1362) by the same group however this report represents the next progression of the approach by combining the screening results of three chemical libraries (13555 compounds). The authors propose that TD-6450 induce stiffening can rapidly move into the clinic as a transmission blocking agent.

While the biomimetic drug screening approach is highly innovative and a technical achievement, the manuscript lacks a sufficient *in vitro* data on TD-6450 activity on early and late gametocytes as well as its effect on developing oocysts in the mosquito. The retention/stiffening effect alone of a potential transmission blocking drug maybe insufficient to warrant its further development especially with PFATP4 inhibitors in development that have both a killing and retention effect. TD-6450 has moderate activity but more studies are needed to characterize its antimalarial activity. Any additional supporting studies to establish the gametocytocidal activity would strengthen the case for the further development of TD-6450 as a transmission blocking drug.

Overall comments/questions.

1. One of the confirmed hits were identified as HDAC inhibitors (9 hits) which have known activity on multiple parasite life stages. The authors should elaborate why this class which has both potent killing and a retention effect was not further investigated. Presumably TD-6450 and L-THP were selected primarily for their potent retention effect, but it is not clear to me why the authors chose to deprioritize the 30% of hits from the ReFrame library that had both killing and retention effects.

2. The authors should clarify the killing activity reported in Figure 3. Is this activity on sexual stages? Have the compounds been tested for their activity on both early and late-stage gametocytes?

3. The authors make a case for the further development of TD-6450 as a transmission blocking agent based on its safety profile, PK and prior use in patients. However, I think the manuscript would be strengthened by some additional studies towards better understanding the gametocytocidal activity of TD-6450. The compound displays moderate activity on asexual and sexual stages. Have the authors considered other in vitro models to measure the transmission blocking efficacy of TD-6450? Many candidate compounds being developed as transmission blocking agents (including NITD609) have been evaluated in the Standard Membrane Feeding Assay (SMFA). TD-6450 has gametocytocidal activity and should show an effect in the SMFA. Is there also a way to assess the retention effect of TD-645 in vivo? Malaria infected mice treated with TD-6450 should have reduced numbers of circulating late-stage gametocytes. Would this be reflected in the oocyst count of mosquitoes fed on TD-6450-treated mice? These studies can also provide insight into establishing the efficacious dose.

4. Is the stiffening activity (retention effect) of TD-6450 representative of this class of molecules? In the case NITD609, the phenotype shown in Figure 5A is representative of the spiroindolones class of molecules. Without preparing and testing close analogs of TD-6450, it is currently unclear if this is the effect of a singleton or more representative of a class of compounds. Although the authors have identified other close analogs of TD-6450 in Table S3, they do not report any antimalarial activity. It would be important to understand if these related compounds also induce stiffening and/or have antimalarial activity. If this is a class effect like the spiroindolones, it may imply that the retention and antimalarial activity could be modulated and perhaps improved.

5. Line 340. Can the authors clarify if the 14 other NS5A inhibitors in the ReFrame library are a different chemical class to TD-6450?

6. Figure 2 panel B. The authors should clarify the absolute stereochemistry of TD-6450. As currently denoted, the four stereocenters are not defined and the left-hand side of the structure is truncated.

The authors claim that a transmission blocking agent does not need to also be an effective antimalarial, however when compared to PfATP4 inhibitors in development, is it difficult to rationalize the further development of a class of transmission blocking agents without antimalarial activity. I would recommend the authors include additional transmission blocking studies on the parasite (e.g., gametocytocidal activity and/or SMFA) before accepting the manuscript.

Reviewer #2 (Remarks to the Author):

Nature Comms Review August 2022

This manuscript describes a large systematic screen using spleen-mimicking microfiltration to identify compounds that increase the stiffness of Plasmodium falciparum gametocytes. This is a hugely technically challenging project – P. falciparum gametocytes are notoriously difficult to culture reproducibly, particularly at the scale needed here, and the adaptation of the bead-based assay to high throughput screening with good z' scores is to be applauded. The outcomes are generally clearly explained (with some exceptions, noted below), and the data convincing. Several promising hits are identified, and two followed up in more detail, in part using data from a previous clinical trial for Hepatitis C. There are key questions that are not addressed in the manuscript, specifically the mechanism of action of the lead compounds, and whether their gametocyte stiffening effect actually leads to a decrease in transmission (and how/if that can be completely disentangled from a gametocyte-killing effect), but it is reasonable that those are topics for future work, rather than being addressed here in a manuscript that already includes a very significant amount of data. Overall this is a comprehensive and highly novel piece of work that will be of broad interest. Some areas need clarification/further explanation however.

Major issues

1. Clarification of compound selection/prioritisation. How different compounds from the three different libraries moved through the screens is not always clearly explained, either in the text (lines 118-43) or in Figure 1. The text is confusing in part because it jumps back and forth between the three different libraries – it might be simpler to break this section into three short paragraphs, one summarising the results from each screening stage for one specific library, rather than trying to combine them all into one rather confusing paragraph. In Figure 1, the different colours of shading/circles are not clearly explained, and it is quite convoluted to trace the +17 +63 +2 notations back to the text to understand what they refer to. Separating all three libraries might also be helpful in this Figure. Whatever approach is used, this section and figure need some significant reworking for clarity.

2. Dose Response Analysis. The initial screen was carried out at a single concentration, 1.11uM, then hits identified from linear regression were used in dose-response assays to prioritise them further. What is not explained however is how the DRA was used as a screen – was this simply a test to see whether the effect (either killing or stiffening) was dose-dependent? Or was there a specific IC50 cut-off? And was a compound viewed as positive for the DRA screen if it had either a killing effect or a stiffening effect, or did all positives have to have a stiffening effect (either with or without a killing effect)?

3. Hepatitis C clinical trial data. A significant part of the data in the manuscript comes from a previous clinical trial using TD-6450 for Hepatitis C treatment, but the relationship between that work and this is never clearly explained. Has that trial ever previously been published? If not (and the clinical trial ID reference at clinicaltrials.gov does not have any data posted), then this manuscript would be the first publication of data from that trial. If so, it would be more general practice to include some more information about the purpose and process of the trial, rather than simply referring to the trial ID. Does the data in Table 2 come from that trial? Presumably so, but it is not actually referenced as such in the Table legend or text. Much better clarification of the status of that trial, and how the data was obtained, is needed.

4. Mechanism of action. While as noted above, it would be unreasonable to expect highly detailed mechanism of action studies in a manuscript such as this which reports a very large screen, given that the discovery of the stiffening impact of TD-6540 in particular is the major novel hit, it would be useful to have at least some indication of what is happening in treated gametocytes. Figure 4 simply shows that it does not cause circularisation like NITD609 does, but is there any morphological impact at all, or any impact on gametocyte development? Is the expression of any well-studied markers impacted by TD-6540 treatment?

Minor issues

1. Serum. Does the use of heat-inactivated serum in parasite culture cause any issues of variability, because presumably multiple different sources of serum had to be used over the course of the large screen? Or were large pooled batches generated and used consistently in all screens?

2. Exposure. How long were the parasites exposed to compound-containing medium in the primary screen or DRA screen before the microsphiltration assay? This doesn't seem to be stated in the Methods, apologies if I've missed it...

3. Imaging flow cytometry. The broad interpretation of the flow imaging plot in Figure 4 is clear – there is a rightward shift in the NITD609 treated gametocytes, which corresponds with the circularisation visible by microscopy. However, the details are missing both in the legend and methods – what exactly is

aspect ratio measuring, how does it correspond to circularity, and is this a statistically significant shift or not?

4. Title. Do the authors really know that the 'safe drugs' identified block transmission, as the title states? This data (and other data, in the case of NITD609) show clearly that they kill gametocytes and increase gametocyte stiffening, but has an actual impact on transmission been measured in both cases? If not, the title is not strictly accurate – these are definitely exciting transmission blocking candidates, but not yet proven to block transmission.

Reviewer #3 (Remarks to the Author):

The manuscript entitled “Safe drugs to block the transmission of malaria revealed by a spleen-mimetic screening approach” by Carucci et al. is a follow-up on previous research conducted by the laboratory of Pierre Buffet. It leverages an innovative technological platform that attempts to recapitulate in vitro the natural spleen quality-control process that eliminates Red Blood Cells (RBC) that have abnormal flexibility/deformability. Plasmodium infected RBC (iRBC) are similarly removed by the spleen, and it has been shown that antimalarial drugs treatments may accelerate this process by inducing an additional stiffening of the iRBC. Here the authors have miniaturized this platform to enable the screen of small compounds libraries with the goal to identify drug leads/candidates that could be combined with other antimalarial agents. By specifically increasing the rate of elimination of Plasmodium gametocytes, which are the sexual forms of the parasite taken up by the Anopheles mosquito vectors, these compounds could block malaria transmission. Reducing malaria transmission is a highly desirable property for all novel therapeutic interventions, and this work is thus a very valuable scientific contribution in the global war on malaria.

The authors first describe the results of the screening efforts and robustly validate the methods employed with the “re-discovery” in the primary screen of compounds known to kill Plasmodium gametocytes and enhance spleen clearance through the stiffening of iRBC (e.g. spiroindolone NITD609, and other inhibitors of PfATP4). The authors subsequently establish a robust hits-list followed with the appropriate characterization in dose response assays and with some orthogonal approaches. They then focus the manuscript on a more in-depth pharmacological characterization of the transmission blocking potential of the drug candidate TD-6450, an HCV NS5A inhibitor not previously known for its antimalarial activity.

The title of the manuscript is not well supported by the data presented in the paper, when it comes to asserting the identification of safe drugs blocking transmission of malaria in the clinics, one would need

a lot more data than what is presented here. One could perhaps consider a title a little less ambitious and more prudent. Nonetheless, in totality, these findings are compelling and should be published. Below are several suggestions for improvements that the editors and authors should consider before publication.

Suggestions for improvement:

1. The primary screen of very well characterized libraries that have been screened in number of malaria assays offers the opportunity for the readers to appreciate the specificity of the compounds in the microfiltration assay. The authors should provide some ancillary data in Table S1 for all primary hits' activity in Plasmodium replication assays in asexual blood (and liver stages?) as well as some standard cellular cytotoxicity assays on mammalian cell lines. This data should be relatively readily available and would increase the significance of the results reported.
2. The authors should strengthen their description of the criteria (i.e. % inhibition in killing and/or retention activity) used for hit selection at the screening concentration which should be clearly stated. This reviewer could not readily find this information in the manuscript current form, and this must be provided to understand the rationale and statistical framework used by the authors to progress compounds to DRA and further characterization.
3. NITD609 (aka KAE609 and cipargamin) anti-gametocidal activity and effect on the rheology of the RBC has been previously reported and thus KAE609 was proposed to be a potent transmission blocking antimalarial drug candidate. It is thus very reassuring to see this compound coming out as a top hit of the primary screen. The authors have very interestingly decoupled the killing activity of the compound from the retention activity in the microfiltration device which should allow to dissect these two distinct pharmacological activities especially when we consider the lack of killing activity of TD-6450 (see below point). Have the authors considered testing some of the KAE609 drug resistant mutants in their assays? Would they expect to see an impact of the drug resistance mutations solely on killing activity or also in retention activity like reported in Zhang et al. 2016?
4. The justification for the selection of only TD-6450 for more in-depth pharmacological analysis is a little perfunctory and few thoughts are given on how the data reported here for cipargamin may inform its further clinical development. The authors should consider running for cipargamin an analysis similar than that of TD-6450 since all the PK/PD phase I and phase IIa/b data are reported and available for cipargamin.

5. TD-6450 antimalarial property is largely unexpected given the origin of the compound as an antiviral drug with a target not expected to be present in Plasmodium. Its transmission blocking activity seems to be primarily driven by the activity on the iRBC deformability. The authors should clarify whether the transmission blocking activity of the compound is related solely to the retention activity given the moderate activity reported on the asexual stages of the parasite. Have the authors measured retention activity of TD-6450 in uninfected RBC? If the target of the TD-6450 transmission blocking activity in the RBC itself? This would offer some opportunity since one would expect a very high barrier to drug resistance but also some challenges as one might expect some on target toxicology. This is an important point that merits some mention at least in the discussion.

6. The moderate activity of TD-6450 in the asexual stages assay offered the opportunity to conduct some drug resistance studies that yielded some reagents to further explore whether the effect on retention of the iRBC is truly decoupled from the antimalarial activity. Have the authors considered running microfiltration experiments with TD-6450 drug resistant mutants? Have the authors sequenced the genomes of those mutants? Have they identified significant single nucleotide polymorphisms? Please clarify.

7. The authors have carefully repeated the experiments to ensure adequate representation of the variability of the data (Fig. 4, 5 and 7) but the reviewer was not able to determine whether the replicates were technical or biological and this should be clarified to the reader.

Minor comments:

1. Fig 1. KDU731 is a known antimalarial drug candidate and a lipid kinase (PI4K) inhibitor and is not appropriately classified in drug families (ie it is not a phosphatase inhibitor). Please describe selection criteria and screening concentration on figure and/or legend.
2. Table 1 MMV390048 is also a PI4K inhibitor with a large volume of PK/PD available. Why did the authors deprioritize further analysis for this compound? Was this solely on PK data? This might be worth being more explicit about this given the interest for this drug target.
3. Fig. 2. Why did compounds fail to reconfirm (blue empty triangles)? Was this after re-synthesis of the compounds or because the further biological replicates were conducted and failed?
4. Please clarify the meaning of “not interpretable” on line 128. No data or data that do not pass QC?
5. Line 129. Please explain the impact of the technical challenges of the assay on the overall screen. Do we expect high number of false positives or are we missing a lot of false negatives?
6. Line 175-176 please explain why the authors did not follow-up on L-THP?
7. Table 2 could be considered as supplemental if data is available online in another format.

8. Line 187 has a typo (activities)

9. Line 199 Provide references for “known parasite-swelling”.

10. Line 204-205 NITD609 is not preferentially active on male gametocytes.

11. Line 256-258 efforts should be made to clarify the rationale for the selection of a 200 nM threshold and consider the impact of this prediction on a potential novel drug combination drugs regimen. How many days would a patient need to take the drug to significantly impact malaria transmission? 1000 mg is a relatively large dose, and this could also be a consideration for a suitable drug partner for a combination?

12. Efforts should be made to make the discussion more concise (lots of repetitions of the results) and discuss more transparently the opportunity and challenges of the proposed approach to block malaria transmission.

13. Line 335 the word “enzymes” is a poor generic descriptor of PfATP4 (consider channels or ion transporters).

Dear Editor,

Answer to reviewers comments

Reviewer #1 (Remarks to the Author):

Targeting late-stage gametocytes and screening large chemical libraries to identify potential transmission blocking (TB) drugs have been widely reported, but this is the first example of a spleen-mimetic being used to screen more than 12000 compounds from the ReFrame drug repurposing library. The microfiltration approach is inventive and a technical achievement that has been previously described in detail (Methods Mol. Biol. 2013; 923:291 and Nat. Protoc. 2018 13:1362) by the same group however this report represents the next progression of the approach by combining the screening results of three chemical libraries (13555 compounds). While the biomimetic drug screening approach is highly innovative and a technical achievement, the manuscript lacks a sufficient in vitro data on TD-6450 activity on early and late gametocytes as well as its effect on developing oocysts in the mosquito. The retention/stiffening effect alone of a potential transmission blocking drug maybe insufficient to warrant its further development especially with PFA IP4 inhibitors in development that have both a killing and retention effect. TD-6450 has moderate activity but more studies are needed to characterize its antimalarial activity. Any additional supporting studies to establish the gametocytocidal activity would strengthen the case for the further development of TD-6450 as a transmission blocking drug.

We thank the reviewer for a clear expression of perceived strengths and weaknesses of the manuscript. Issues raised in this general statement are carefully addressed point-by-point below.

Overall comments/questions.

1. One of the confirmed hits were identified as HDAC inhibitors (9 hits) which have known activity on multiple parasite life stages. The authors should elaborate why this class which has both potent killing and a retention effect was not further investigated. Presumably TD-6450 and L-THP were selected primarily for their potent retention effect, but it is not clear to me why the authors chose to deprioritize the 30% of hits from the ReFrame library that had both killing and retention effects.

We share the reviewer's opinion that HDAC inhibitors are candidates for further investigations. Histone de-acetylation is involved in *P. falciparum* gametocytes development (Andrews et al, Curr Pharm Des, 2012), making this pathway a potential target for transmission-blocking new anti-malarials. However, we did not prioritize further studies of these compounds because the family displays potentially severe side effects, that we consider non-favorable for a malaria transmission-blocking strategy. As representative examples, when administered in Phase I or Phase II trials both abexinostat (Trial ID: NCT00724984) and givinostat hydrochloride (ID: NCT01761292) caused significant adverse events in more than 15% of patients (anemia, abdominal pain, vomiting, pyrexia, decreased platelet count, rhabdomyolysis and hematuria). Others (CUDC-907, ID: NCT02674750, and AR-42 ID: NCT02795819) were associated with a > 25% rate of life-threatening adverse events, although the severity of the underlying condition may have contributed to their emergence. We have made these points more explicit lines 323-325 (now line 343), replacing the previous text: by "*We identified a cluster of 9 histone deacetylase inhibitors (HDACs) which are under clinical development for cancer, and*

*promising candidates for other diseases, including symptomatic malaria⁵¹, providing further evidence that the HDAC pathway is involved in malaria transmission⁵². However, we did not explore this family further because after a thorough analysis of their safety, we found it acceptable to treat cancer or potentially severe symptomatic malaria, but probably not for malaria transmission-blocking interventions^{53,54}. Future optimizations may identify HDACs inhibitors more selective for *P. falciparum* and with a better safety profile⁵⁵”.* The safety scoring in Table 1 is based on a systematic analysis of all available published and unpublished safety information, performed for all selected hits, a process that took 2 months of full-time work.

Regarding our choice to focus post-screening exploration on compounds with a predominant stiffening effect, we assume that the killing effect (including that of HDAC inhibitors) has generally been observed by previous screening campaigns, resulting in detailed explorations by other teams. We therefore prioritized the very demanding post-screening explorations on best candidates displaying the most original effect captured by our screening campaign (i.e., stiffening).

2. The authors should clarify the killing activity reported in Figure 3. Is this activity on sexual stages? Have the compounds been tested for their activity on both early and late-stage gametocytes?

The killing activity reported in the Figure 3 has been determined on mature stage V gametocytes only. Being the only circulating transmission forms, deformable mature stage V gametocytes were our cellular target. *P. falciparum* immature gametocytes are rigid, which probably explains why they are never seen in the circulating blood of subjects with a normal spleen function, and observed exclusively in organs (bone marrow, spleen), and sometimes in the circulating blood of splenectomized or hyposplenic subjects (Henry et al. Trends Parasitol 2021). Because they are naturally rigid, immature gametocytes are not amenable to a deformability-based screen. They are indeed spontaneously retained in microfilters, even before being exposed to compounds (Tiburcio et al. Blood 2012). As expressed under Point 1 above, we assume that compounds with a marked killing activity have been captured by previous killing-based screen campaigns targeting gametocytes. This was the case for NITD609 but not for TD-6450, and more generally NS5A inhibitors. It is therefore on purpose that we did not test compounds on immature gametocytes. As explained in more details under Point 3 below, killing activity of confirmed hits against early-stage gametocytes would only mildly reinforce their potential clinical impact. If the clearing effects of NITD609 and TD-6450 are confirmed in humans, we will obtain additional resources necessary to explore this side dimension of the approach.

3. The authors make a case for the further development of TD-6450 as a transmission blocking agent based on its safety profile, PK and prior use in patients. However, I think the manuscript would be strengthened by some additional studies towards better understanding the gametocytocidal activity of TD-6450. The compound displays moderate activity on asexual and sexual stages. Have the authors considered other in vitro models to measure the transmission blocking efficacy of TD-6450? Many candidate compounds being developed as transmission blocking agents (including NITD609) have been evaluated in the Standard Membrane Feeding Assay (SMFA). TD-6450 has gametocytocidal activity and should show an effect in the SMFA. Is there also a way to assess the retention effect of TD-645 in vivo? Malaria infected mice treated with TD-6450 should have reduced numbers of circulating late-stage gametocytes. Would this be reflected in the oocyst count of mosquitoes fed on TD-6450-treated mice? These studies can also provide insight into establishing the efficacious dose.

We apologize for a quite long (though hopefully helpful) answer to these major queries.

Regarding mechanisms of action, cellular impacts of NITD609 and TD-6450 on RBC harboring mature gametocytes, we have shown by imaging flow cytometry that NITD609 induces swelling whereas TD-6450 does not markedly alter cell shape (please see also the detailed answer to Point 4 of Reviewer 2) indicating different modes of action. We agree that fine cellular and molecular mechanisms by which TD-6450 alters gametocyte deformability will deserve further explorations in the future.

Regarding the potential impact of complementary explorations *in vitro*, we agree with the reviewer that all relevant information must be obtained before engaging into clinical development. However, in the specific context of our approach, SFMA and studies in mice are dispensable prerequisites, as supported by the following considerations

Physiology & Parasitology: Clinical and experimental observations in human subjects either healthy, affected by hereditary spherocytosis, or experimentally transfused with RBC show that the spleen rapidly clears altered RBC (Jandl & Castle, JCI 1957, Roussel et al. Blood 2021). Based on *ex-vivo* perfusion of human spleens (Buffet et al. Blood 2006, Safeukui et al. Blood 2008), as well as ektacytometry, imaging flow cytometry, and microsphiltration used here for screening (Deplaine et al. Blood 2011, Safeukui et al. Blood 2012, Safeukui et al. PLoS One 2013) we had confirmed previous seminal observations (Cranston et al. Science 1984) that RBC retention applies to malaria, including mildly altered RBC infected by young asexual forms (rings). More than a decade ago, we predicted that the human spleen should retain and concentrate *P. falciparum* ring-infected RBC (Buffet et al. Blood 2011), a prediction directly confirmed by a recent splenectomy study in a malaria-endemic area (Kho et al. N Engl J Med & PLoS Med 2021). **Our prediction that drug-induced stiffening of mature gametocytes will clear them from the circulation is based on this very solid framework.**

Parasitology and cell biology: **Terminally differentiated gametocytes do not multiply.** Inducing retention is therefore enough for full drug efficacy, killing is not necessary. Even alive, gametocytes retained in the spleen are inaccessible to the vector. What matters is the intensity of this retention and its sustainability.

Data collected in humans: Studies in *P. falciparum* carriers show that transmission to mosquitoes declines from 70-80% at a gametocyte blood density (gametocytemia) of 250/ μ l through 40% at 100/ μ l to 0% below 30/ μ l (Meibalan et al, J Infect Dis, 2021 and Churcher et al, eLife, 2013). A decrease in gametocytemia is thus directly related to reduced transmission, and a 70-80% retention is likely to be fully effective in a vast majority of subjects. **A major strength of our approach is therefore that simply measuring decrease on gametocytemia is enough to predict the drug impact on transmission.** For drugs acting by killing like primaquine, gametocytemia and transmission are not tightly related as it is not evident to determine viability of detected gametocytes. In that case, confirmation by SMFA is essential, but our approach can skip this demanding step. We will directly determine whether further development is warranted based on the simple measure of gametocytemia after drug administration. The dose will be determined by results from previous trials and by the modelling part of this manuscript (Figs 6 & 7) as data on safety (Table 2) and PK (Table 3) are already available. NITD609 and TD-6450 have demonstrated to be safe so, they can be administered in the context on a Controlled Human Infectious Challenge. Weeks-long presence of circulating mature gametocytes has been recently obtained in piperazine-treated volunteers (Stepniewska et al, J Infect Dis, 2022) opening the way to such studies. Regarding sustainability, a major advantage of TD-6450 is that effect will persist beyond the long exposure to this slowly

eliminated drug. The brief powerful action of NIT 609 and the sustained effect of TD-6450 are expected to synergistically provide the desired effect on transmission.

We have added a few sentences to the discussion to make these points more explicit, line 406: *“Preclinical information very important for the development of anti-gametocytic drugs such as Standard Membrane Feeding Assays and studies in mice are dispensable prerequisites in the specific context of our approach. A decrease in gametocytemia is indeed directly related to reduced transmission, and we provide very robust results suggesting that TD-6450 and NITD609 may reduce gametocytemia.*

Regarding mouse models, sadly the pathophysiological processes under study are not mimicked accurately in animal models, even in “humanized” mice. Mature asexual stages and immature gametocytes (which are both rigid) are observed in the circulation of *P. falciparum*-infected “humanized” mice (Duffier et al. Scientific reports 2016) confirming the inadequate spleen function of these mice with regards to mechanical retention of altered RBC. This is addressed in the manuscript line 400: *“In the absence of a validated animal model for the splenic retention of mature P. falciparum gametocytes⁴⁴,...”*

4. Is the stiffening activity (retention effect) of TD-6450 representative of this class of molecules? In the case NITD609, the phenotype shown in Figure 5A is representative of the spiroindolones class of molecules. Without preparing and testing close analogs of TD-6450, it is currently unclear if this is the effect of a singleton or more representative of a class of compounds. Although the authors have identified other close analogs of TD-6450 in Table S3, they do not report any antimalarial activity. It would be important to understand if these related compounds also induce stiffening and/or have antimalarial activity. If this is a class effect like the spiroindolones, it may imply that the retention and antimalarial activity could be modulated and perhaps improved.

Following the line suggested by the reviewer, we have reanalyzed the retention of the 14 NS5A inhibitors upon primary screening (one well/compound), and compared it with the respective negative controls (DMSO) in each plate. We observed that of these 14 NS5A inhibitors 11 showed a retention value higher than that of the DMSO, and 3 of them display a differential retention rates between 10% and 13%, versus 32% for TD-6450 (raw retention rates were in the range of 40-70% because the retention of DMSO-exposed gametocytes was generally 25%-50%, please see Fig2). Based on this observation, we agree with the reviewer that a class effect cannot be excluded, although TD-6450 is the best candidate, as further supported by the safety and PK data provided in the second part of the manuscript. Further post-screening explorations of the whole NS5A family may indeed generate interesting information in the future. The major fact however is that of the 14 NS5A inhibitors tested, only TD-6450 met the criteria to be considered a hit. We have made the hits selection process from the primary screen more explicit at the “Methods” section as follows (lines 475): *“Only those compounds that fell outside the upper prediction band in both readouts were finally selected as hits.”* We have also mentioned a potential class effect (please see answer to the next point).

5. Line 340. Can the authors clarify if the 14 other NS5A inhibitors in the ReFrame library are a different chemical class to TD-6450?

All 14 compounds are NS5A inhibitors but TD-6450 is the only heterodimer in this family. We have amended the text (lines 367) as follows, *“Of the 14 NS5A inhibitors included in the ReFrame library, only TD-6450 met the hit definition. TD-6450 is the only heterodimeric NS5A*

inhibitor which may explain its enhanced activity. Nevertheless, a class effect cannot be excluded at this stage.”

6. Figure 2 panel B. The authors should clarify the absolute stereochemistry of TD-6450. As currently denoted, the four stereocenters are not defined and the left-hand side of the structure is truncated.

We thank the reviewer for helping us correct these weaknesses of the manuscript. The absolute stereochemistry of TD-6450 is (S)-1-((S)-2-(5-(4'-(6-((2R,5S). It is a pure enantiomer. We replaced the chemical formula of TD-6450 showed in Fig.3:

TD-6450

With the following:

TD-6450

The same has been done for NITD609 and L-THP for homogeneity. The legend of the figure 3 has also been adapted as follows (line 152):

Fig.3. Dose-response curves of selected hits. Dose-response curves and chemical structures of the 3 selected hits: NITD609 (A), TD-6450 (B), and L-THP (C). TD-6450 was a pure enantiomer. Stereocenters are: (S)-1-((S)-2-(5-(4'-(6-((2R,5S).

7. The authors claim that a transmission blocking agent does not need to also be an effective antimalarial, however when compared to PfATP4 inhibitors in development, is it difficult to rationalize the further development of a class of transmission blocking agents without antimalarial activity. I would recommend the authors include additional transmission blocking studies on the parasite (e.g., gametocytocidal activity and/or SMFA) before accepting the manuscript.

We thank the reviewer for his recommendation, but we still robustly rationalize further development of TD-6450 despite a relatively weak killing effect on mature gametocyte, which will likely be similarly weak on immature gametocytes. It took the originality of our spleen-

mimetic approach to uncover NSSA inhibitors as potential important players in malaria where several killing-based campaigns had not selected any member of the family. Therefore, they likely display at best a weak killing effect on immature gametocytes. As explained in detail in our answer to Point 3, what matters is the amplitude and sustainability of the retention effect on mature gametocytes. Our results (Fig 4) show that despite their outstanding qualities, PfATP4 inhibitors will likely display a transient transmission-blocking effect, whereas the stiffening effect of TD-6450 is more sustainable, likely persisting for at least several days. In addition, the PfATP4 inhibitors family is strongly exposed to the risk of drug resistance (Rottmann et al, Science, 2010). Therefore, the synergistic transmission-blocking effect of the combination both drugs (Fig 4), that includes a fast-acting (NITD609) and a slow-acting component (TD-6450) is reminiscent of what ACT have brought to the treatment of malaria attacks, with mutual cross protection against resistance.

Reviewer #2

This manuscript describes a large systematic screen using spleen-mimicking microfiltration to identify compounds that increase the stiffness of Plasmodium falciparum gametocytes. This is a hugely technically challenging project – P. falciparum gametocytes are notoriously difficult to culture reproducibly, particularly at the scale needed here, and the adaptation of the bead-based assay to high throughput screening with good z' scores is to be applauded. The outcomes are generally clearly explained (with some exceptions, noted below), and the data convincing. Several promising hits are identified, and two followed up in more detail, in part using data from a previous clinical trial for Hepatitis C. There are key questions that are not addressed in the manuscript, specifically the mechanism of action of the lead compounds, and whether their gametocyte stiffening effect actually leads to a decrease in transmission (and how/if that can be completely disentangled from a gametocyte-killing effect), but it is reasonable that those are topics for future work, rather than being addressed here in a manuscript that already includes a very significant amount of data. Overall this is a comprehensive and highly novel piece of work that will be of broad interest. Some areas need clarification/further explanation however.

Major issues

1. Clarification of compound selection/prioritisation. How different compounds from the three different libraries moved through the screens is not always clearly explained, either in the text (lines 118-43) or in Figure 1. The text is confusing in part because it jumps back and forth between the three different libraries – it might be simpler to break this section into three short paragraphs, one summarising the results from each screening stage for one specific library, rather than trying to combine them all into one rather confusing paragraph. In Figure 1, the different colours of shading/circles are not clearly explained, and it is quite convoluted to trace the +17 +63 +2 notations back to the text to understand what they refer to. Separating all three libraries might also be helpful in this Figure. Whatever approach is used, this section and figure need some significant reworking for clarity.

Figure 1 and text were modified by splitting the 3 libraries, as wisely advised. Figure 1 and its legend were modified as shown below:

Fig.1. High-throughput screening based on mitochondrial staining and cell deformability identifies compounds with both killing effect and stiffening activity on *P. falciparum* late gametocytes. Screening progression cascade of three different libraries: Malaria Pathogen Box (A), Kinase Inhibitors Box (B), and ReFrame library (C). (A) 3 hits from primary screening were submitted to dose-response analysis along with 12 compounds found active in some but not all screening replicates. The 3 hits were confirmed but none of them was selected for further post-screening validation. (B) 4 hits from primary screening along with 5 compounds found active in some but not all screening replicates were submitted to dose-response analysis raising 3 confirmed hits. None of them was selected for further post-screening validation. (C) 112 hits from primary screening were submitted to dose-response analysis, raising 74 confirmed hits. 63 compounds with uninterpretable results during primary screening were added to the hits for dose-response analysis raising additional 2 confirmed hits. The 76 confirmed hits were allocated to 7 groups (panels on the right), based on their activity and molecular target. For each group, one representative hit has been selected for illustration. Hit scoring based on route of administration, safety in human subjects, and pharmacokinetics resulted in the selection of 3 drugs submitted to final confirmation experiments (dark blue).

Text was modified in the lines 74-75: “the Kinase Inhibitors Box from GSK (400 and 350 compounds, respectively, Fig. 1 A-B), and furthermore the larger ReFrame repurposing library (12,805 compounds, Fig. 1C).”, and lines 101-109: “Primary screening was repeated six or three times for small libraries (Kinase Inhibitors and Pathogen Boxes). Hits were selected plate by plate, based on linear regression of compound distributions on screening results plots (Fig. 2C-D). We found three and four hits respectively in the Pathogen and Kinase Inhibitors Boxes (Fig. 1 A-B). Their respective hit rates were 0.75%, and 1.14%. Compounds active in some but not all screening replicates were added to the hits for further analysis. The larger ReFrame library was screened in singlicate, raising 112 hits (0.87% hit rate, Fig. 1C). The detected hits had either a predominant stiffening activity (44%), a predominant killing effect (28%), or a combination of both (28%, Fig. 2A-B). Z' values were between 0.4 and 0.7 (Fig. S1). These 112 hits, plus 63 compounds (0.6%) with uninterpretable results upon primary screening were explored further.”

2. Dose Response Analysis. The initial screen was carried out at a single concentration, 1.11uM, then hits identified from linear regression were used in dose-response assays to prioritise them further. What is not explained however is how the DRA was used as a screen – was this simply a test to see whether the effect (either killing or stiffening) was dose-dependent? Or was there a specific IC50 cut-off? And was a compound viewed as positive for the DRA screen if it had either a killing effect or a stiffening effect, or did all positives have to have a stiffening effect (either with or without a killing effect)?

The DRA was just used to confirm the effect of hits selected by the primary screening, and broadly estimate their potency by determining their IC50 for killing and stiffening (Fig. 3, Table S1) but no IC50 cut-off value was used for further hit prioritization. Confirmation by DRA of stiffening, killing, or the coexistence of both was enough for hit prioritization. The following sentence was added in the line 121: “119 hits were selected for dose-response analysis (DRA): three from the Pathogen Box, four from the Kinase Inhibitors Box, and 112 from the ReFrame library. Confirmation of the stiffening effect, killing effect or coexistence of both was the criterion for further analysis of hits. IC50 values were obtained for both killing effect and stiffening activity (Fig. 3, Table S1) but no IC50 cut-off value was used for further hit prioritization. Hits were deprioritized when no IC50 could be determined.”

3. Hepatitis C clinical trial data. A significant part of the data in the manuscript comes from a previous clinical trial using ID-6450 for Hepatitis C treatment, but the relationship between that work and this is never clearly explained. Has that trial ever previously been published? If not (and the clinical trial ID reference at clinicaltrials.gov does not have any data posted), then this manuscript would be the first publication of data from that trial. If so, it would be more general practice to include some more information about the purpose and process of the trial, rather than simply referring to the trial ID. Does the data in Table 2 come from that trial? Presumably so, but it is not actually referenced as such in the Table legend or text. Much better clarification of the status of that trial, and how the data was obtained, is needed.

The reviewer is right: the results of the phase I clinical trial were never published, except for a poster, referred as P0898, in the Journal of Hepatology, vol.62, S680-1, 2015. As wisely requested, a detailed paragraph was added at the Methods section before the paragraph entitled “PK modeling” (line 522) to provide information about the protocol, IRB approval and inclusion criteria of the study.

Codice campomodificato

TD-6450 phase I clinical study. This was a phase I, double-blinded, randomized, placebo-Controlled, Single Ascending Dose (SAD) and Multiple Ascending Dose (MAD) study to evaluate the safety, tolerability, pharmacokinetics, and food effect of TD-6450 in healthy subjects (clinicaltrial.gov identifier NCT02022306). The study started on February the 4th 2014 (first subject enrolled) and was concluded on August the 18th 2014 (last subject, last visit) performed by ICON Development Solutions San Antonio, Texas, United States under the sponsorship of Theravance Biopharma. The protocol and all amendments for this study and all accompanying material that was provided to subjects (including advertisements, subject's information sheets, and descriptions of the study used to obtain informed consent) were submitted by the investigator to the centralized Institutional Review Board (IRB), namely the IntegReview, established in 1999 at Austin (Texas, U.S.A.) and acquired by Advarra in 2020. Documented approval was obtained before the study initiation, on March the 28th 2014. This study was conducted in accordance with the protocol, the principles of the International Conference on Harmonization of Technical Requirements for Registration of Pharmaceuticals for Human Use (ICH) Guideline for Good Clinical Practice (GCP), the United States Code of Federal Regulations, the principles of the World Medical Association Declaration of Helsinki, Ethical Principles for Medical Research Involving Human Subjects, and all applicable regulatory requirement.

This study was conducted in 2 parts, Part A (SAD) and Part B (MAD). Part A: Healthy subjects were sequentially enrolled and randomly assigned to each dose cohort (up to 10 dose-ascending cohorts were planned, including a fed arm in the food effect cohort) to receive either a single dose of TD-6450 or placebo. For each dose cohort, 8 subjects were randomized in a 3:1 ratio (6 subjects received TD-6450 and 2 subjects received placebo). The starting dose of TD-6450 was 0.5 mg with a planned maximum dose of 1000 mg. Upon review of the 500-mg cohort PK data, the decision was made to not escalate to the 1000-mg cohort because of exposure saturation at ≥ 500 mg. The following doses were administered: 0.5 mg, 1.5 mg, 5 mg, 15 mg, 30 mg, 60 mg, 120 mg, 240 mg, and 500 mg. Part B: Healthy subjects were sequentially enrolled in 3 ascending dose cohorts (TD-6450 60 mg, 120 mg, and 240 mg) and randomly assigned to receive either TD-6450 or placebo as once daily doses for 14 days. For each dose cohort, 10 subjects were randomized in a 4:1 ratio (8 subjects received TD-6450 and 2 subjects received placebo). Blinded safety, tolerability, and available PK data after each dose cohort were reviewed by the Safety Data Review Committee (SDRC) before escalation to the next dosing cohort for both part A and B.

A total of 81 subjects were enrolled in Part A and 30 subjects were enrolled in Part B; all subjects were included in the safety analyses and all subjects who completed the study were included in the PK analyses. At screening, eligible subjects were 18 to 60 years of age (inclusive), had a body mass index 18 to 30 kg/m² (inclusive), weighed at least 50 kg, and were in good health as judged by the absence of clinically significant diseases or clinically significant abnormal laboratory values (Screening or Day -1).

The sentence at Table 2 footnotes (line 192) “(administered once daily during 14 days)” was deleted. The text has been also modified adding more information about the purpose and the process of the trial. The following sentence was added at the beginning of the paragraph “Pharmacokinetics of TD-6450” (line 250): “TD-6450 was discovered and developed for the treatment of Hepatitis C virus infection by Theravance Biopharma Inc, but stopped after phase II for strategic reasons. The first phase I clinical trial was completed in 2014 (clinical trials ID: NCT02022306). The primary objective of this study was to evaluate the safety and tolerability of single ascending dose (SAD) and multiple ascending dose (MAD) in healthy subjects (Table 2). The secondary objective was to determine the pharmacokinetics (PK) of TD-6450 in healthy subjects for both SAD and MAD. The PK data obtained from this phase I study were used to make a concentration-time modelling.”

4. Mechanism of action. While as noted above, it would be unreasonable to expect highly detailed mechanism of action studies in a manuscript such as this which reports a very large screen, given that the discovery of the stiffening impact of TD-6540 in particular is the major novel hit, it would be useful to have at least some indication of what is happening in treated gametocytes. Figure 4 simply shows that it does not cause circularisation like NTID609 does, but is there any morphological impact at all, or any impact on gametocyte development? Is the expression of any well-studied markers impacted by TD-6540 treatment?

No differences could be noted under the microscope between gametocytes treated with DMSO and gametocytes treated with TD-6450. To confirm this visual assessment, we used the feature finder wizard of the IDEAS software 6.2 to compare shape and size parameters between the 2 gametocytes populations. The feature finder wizard provides the top-ranking features that differs between 2 populations and list them in a table with their category and channel. A Statistics table is added to the analysis area that lists the features with the RD Mean for the truth populations. RD is the Fischer's discriminant ratio which is the difference in the means divided by the sum of the standard deviations for the two populations. The larger the RD value, the better the separation afforded by the feature (a $RD > 2$ allows a good separation of 2 populations based on the feature).

Feature finder found no significant differences between the 2 populations, as the highest RD mean was 0.07 a very low value (top ranking feature and RD mean are listed in the table below)

Width_Object(M01,Ch01,Tight)	0.07
Minor Axis_Object(M01,Ch01,Tight)	0.07
Aspect Ratio_Object(M01,Ch01,Tight)	0.04
Symmetry 2_Object(M01,Ch01,Tight)_Ch01	0.04
Symmetry 4_Object(M01,Ch01,Tight)_Ch01	0.04

Minor issues

1. Serum. Does the use of heat-inactivated serum in parasite culture cause any issues of variability, because presumably multiple different sources of serum had to be used over the course of the large screen? Or were large pooled batches generated and used consistently in all screens?

Human serum was provided in bags of approximately 250 ml. 5 to 10 bags were pooled to prepare a single batch. A single gametocytes induction requires 200 ml of medium per day (corresponding to 20 ml of human serum), 15 days to complete the induction. 8 inductions were used to screen the entire ReFrame library. It was therefore impossible to use one single batch for the whole screening campaign. A variability issue related to the different serum batches (among other factors like inter-inductions or inter-plates variability) could not be excluded. For this reason, the robustness of this assay was guaranteed by plate-by-plate analysis with the presence of the positive and negative controls in each plate.

Text has been modified in the Methods section adding this sentence in the line 441:

“Each batch of serum was obtained by pooling 10 serum bags from different donors. Serum was kept at -20°C and thawed in a warmed water bath before use. A new serum batch was prepared when the previous was completed.”

2. Exposure. How long were the parasites exposed to compound-containing medium in the primary screen or DRA screen before the microfiltration assay? This doesn't seem to be stated in the Methods, apologies if I've missed it...

As briefly stated at the Methods section (line 500) and at the Results section, parasites were exposed during 24 hours for both primary screening and DRA. A sentence was added (line 439) to specify that 24-hour was the incubation time for the screening campaign, as follows: *“and then incubated 24 hours both for the screening campaign and the dose-response analyses.”* Thank you for accurately pointing this missing item.

3. Imaging flow cytometry. The broad interpretation of the flow imaging plot in Figure 4 is clear – there is a rightward shift in the NITD609 treated gametocytes, which corresponds with the circularisation visible by microscopy. However, the details are missing both in the legend and methods – what exactly is aspect ratio measuring, how does it correspond to circularity, and is this a statistically significant shift or not?

Aspect ratio is the measure of the minor axis divided by the major axis and describes how round or oblong an object is. One experiment was performed to assess the circularization of gametocytes exposed to DMSO (negative control), NITD609 and TD-6450. The aspect/ratio value is determined for each cell. Mean values were: DMSO = 0.68, TD-6450 = 0.7, NITD609 = 0.76. The difference between DMSO- and NITD-exposed gametocytes was significant ($p < 0.0001$, for DMSO $n = 5479$ and for NITD609 $n = 9006$, where n is the number of events analyzed).

We changed the text at lines 216-219 as follows:

(A) Giemsa-stained erythrocytes infected by a mature gametocyte of P. falciparum exposed for 24 hours to DMSO (negative control, green border), TD-6450 5 μ M (blue border) and

NITD609 1 μ M (red border). Circularity of gametocytes by imaging flow cytometry was compared measuring the aspect ratio showing a significant difference between DMSO and NITD609.

And the following sentence was added in the line 510:

“The aspect ratio is the ratio of the minor axis divided by the major axis and describes how round or oblong an object is. Focused cells and single cells were...”.

4. Title. Do the authors really know that the ‘safe drugs’ identified block transmission, as the title states? This data (and other data, in the case of NITD609) show clearly that they kill gametocytes and increase gametocyte stiffening, but has an actual impact on transmission been measured in both cases? If not, the title is not strictly accurate – these are definitely exciting transmission blocking candidates, but not yet proven to block transmission.

The title was changed to a wiser version as follows: *“Safe drugs with high potential to block the transmission of malaria revealed by a spleen-mimetic screening approach”.*

Reviewer #3 (Remarks to the Author):

The manuscript entitled “Safe drugs to block the transmission of malaria revealed by a spleen-mimetic screening approach” by Carucci et al. is a follow-up on previous research conducted by the laboratory of Pierre Buffet. It leverages an innovative technological platform that attempts to recapitulate in vitro the natural spleen quality-control process that eliminates Red Blood Cells (RBC) that have abnormal flexibility/deformability. Plasmodium infected RBC (iRBC) are similarly removed by the spleen, and it has been shown that antimalarial drugs treatments may accelerate this process by inducing an additional stiffening of the iRBC. Here the authors have miniaturized this platform to enable the screen of small compounds libraries with the goal to identify drug leads/candidates that could be combined with other antimalarial agents. By specifically increasing the rate of elimination of Plasmodium gametocytes, which are the sexual forms of the parasite taken up by the Anopheles mosquito vectors, these compounds could block malaria transmission. Reducing malaria transmission is a highly desirable property for all novel therapeutic interventions, and this work is thus a very valuable scientific contribution in the global war on malaria.

The authors first describe the results of the screening efforts and robustly validate the methods employed with the “re-discovery” in the primary screen of compounds known to kill Plasmodium gametocytes and enhance spleen clearance through the stiffening of iRBC (e.g. spiroindolone NITD609, and other inhibitors of PfATP4). The authors subsequently establish a robust hits-list followed with the appropriate characterization in dose response assays and with some orthogonal approaches. They then focus the manuscript on a more in-depth pharmacological characterization of the transmission blocking potential of the drug candidate TD-6450, an HCV NS5A inhibitor not previously known for its antimalarial activity.

The title of the manuscript is not well supported by the data presented in the paper, when it comes to asserting the identification of safe drugs blocking transmission of malaria in the clinics, one would need a lot more data than what is presented here. One could perhaps consider a title a little less ambitious and more prudent. Nonetheless, in totality, these findings are compelling and should be published. Below are several suggestions for improvements that the editors and authors should consider before publication.

We agree with this wise statement (also expressed by reviewer 2) and have changed the title to: “Safe drugs with high potential to block the transmission of malaria revealed by a spleen-mimetic screening approach”.

Suggestions for improvement:

1. The primary screen of very well characterized libraries that have been screened in number of malaria assays offers the opportunity for the readers to appreciate the specificity of the compounds in the microsphiltration assay. The authors should provide some ancillary data in Table S1 for all primary hits’s activity in Plasmodium replication assays in asexual blood (and liver stages?) as well as some standard cellular cytotoxicity assays on mammalian cell lines. This data should be relatively readily available and would increase the significance of the results reported.

Ancillary data were added to Table S1, as follows:

Table S1. ReFrame library hitlist with chemical structures and IC₅₀ for both killing effect and stiffening activity (columns 3 & 4). Columns 7 & 8 show results from previously published reports on the replication of Plasmodium (72-hour *P. falciparum* Dd2 SybrGreen Protein Binding Fold Shift (PBFS) with 50% serum assay) and cytotoxicity on eucaryotic cells (HEK293T and HepG2 72-hour Cytotoxicity). IC₅₀ are indicated when the compound was selected by our primary screen.

Name	Chemical structure	Killing IC ₅₀ (μM)	Stiffening IC ₅₀ (μM)	Group	Molecular target	“72-h Dd2-SybrGreen PBFS” assay IC ₅₀ (μM)	Cellular cytotoxicity assays IC ₅₀ (μM) (HEK293T & HepG2)
Atiprimodimaleate		6.67	N/A	Kinase and phosphatase inhibitors	Human PKB/Akt	NC	NC
Decamethoxine		8.1	1	Antibiotics & antivirals	Unknown	0.061	3.3 & 4.3
Oligomycin A		N/A	0.5	Antibiotics & antivirals	Human HIF-1	NC	NC
Acetomeroctol		7.2	5	Antibiotics & antivirals	Unknown	NC	0.462 & 1.92

KF 66854		0.76	3.6	Others	5-HT4 receptor	1.64	NC & 5.08
Potassium antimonyl tartrate		3.63	10.7	Others	Unknown	NC	NC
Bortezomib		2.06	N/A	Others	Human Proteasome subunit beta type-5 & 1	NC	NC
Ammonium trichlorotellurate		13.7	2.2	Antibiotics & antivirals	Unknown	NC	NC
Methylthioninium chloride (Methylene Blue)		4.85	2.8	MAO inhibitors	Human guanylate cyclase & nitric oxide synthase	NC	NC
Quisinostat		5.2	3.5	Anti-cancer: HDAC inhibitors	Human HDAC	NC	NC
Gramicidin		0.008	0.07	Antibiotics & antivirals	Bacterial membranes	NC	NC
Abexinostat		2	0.64	Anti-cancer: HDAC inhibitors	Human HDAC	NC	NC
Eseroline		1.565	2.3	Others	Human AcHEIs	NC	NC

CRA-026440		16	0.55	Anti-cancer: HDAC inhibitors	Human HDAC	0.063	0.081 & 0.011
Alexidine dihydrochloride		41	0.55	Antibiotics & antivirals	Unknown	0.032	3.33 & 3.06
Leuco methylthioninium salt (Methylene Blue salt)		23.25	N/A	MAO inhibitors	Human guanylate cyclase & nitric oxide synthase	NC	NC
N-tert-butylisoquinoline		9.55	9.45	Antimalarial agents	Pf Hemoglobin degradation	NC	NC
Unidentified compound		3.3	5.04	Others	Unknown	NC	NC
Romidepsin		20	5	Anti-cancer: HDAC inhibitors	Human HDAC	NC	NC
Bisantrene HCl		2.2	N/A	Others	Human DNA topoisomerase II	0.065	0.054 & 0.062
Auranofin		2.575	3.5	Others	Human TrxR	NC	NC

NVP-BGT226		0.013	1.6	Kinase and phosphatase inhibitors	Human PI3K	NC	NC
Oligomycin B		5.8	1.55	Antibiotics & antivirals	Human HIF-1	NC	NC
Homoharringtonine		1.73	2.3	Others	Human Stat3	0.007	0.032 & 0.122
DDD498		1.7	2.6	Antimalarial agents	Pf EF2	0.004	NC
Unidentified compound		N/A	0.2	Others	Unknown	NC	NC
Unidentified compound		7.45	1.75	Others	Unknown	NC	NC
NITD609		0.15	0.11	Antimalarial agents	Pf ATPase 4	NC	NC
Bismuth ethanedithiol		1.3	1.5	Antibiotics & antivirals	Unknown	NC	NC

YM 161514		5.55	3.6	Others	Human Beta-1 adrenergic receptor	2.26	NC & 3.97
SR-26050		2.6	2.9	Others	Unknown	NC	NC
PPA904		2.415	5.05	Others	Unknown	NC	NC
Tyrothricin		0.29	5.7	Antibiotics & antivirals	Bacterial membranes	0.033	1.8 & 1.9
Pirtenidine		2.785	3.65	Antibiotics & antivirals	Unknown	0.615	1.48 & 0.285
TD-6450		N/A	0.55	Antibiotics & antivirals	HCV NS5A	NC	NC
Cephaeline		1	3.8	Others	Human 5-HT4 receptor	0.028	0.014 & 0.062
BRD-7929		2.2	3.8	Antimalarial agents	Phenylalanine tRNA synthetase	NC	NC
KDU731		0.09	0.12	Kinase and phosphatase inhibitors	Cp PI4K	NC	NC

VE-822		0.71	9.5	Kinase and phosphatase inhibitors	Human ATR kinase	NC	NC
Pyrithione Zinc		2.5	6.2	Antibiotics & antivirals	Fungal proton pumps	NC	NC
PA92		1.15	0.6	Antimalarial agents	Pf ATPase 4	NC	NC
Bispyrithione		0.92	2.35	Antibiotics & antivirals	Fungal proton pumps	NC	NC
AR-42		1.5	1.9	Anti-cancer: HDAC inhibitors	Human HDAC	NC	NC
BN-82685		2.8	4.6	Kinase and phosphatase inhibitors	Human CDC25 phosphatase	1.74	2.92 & 1.89
Paranyline		1.9	3.2	Others	Unknown	1.87	1.42 & 0.65
APPCL		1.2	1.03	Others	Unknown	1.06	0.923 & 0.354
Bruceantin		0.16	0.02	Others	Unknown	NC	NC
Sepantronium bromide		1.35	1.89	Others	Unknown	NC	NC

Ceritinib		10	10.5	Kinase and phosphatase inhibitors	Human ALK1	NC	NC
Chlorproguanil hydrochloride		N/A	10.35	Antimalarial agents	Pf antifolate	NC	NC
Daunorubicin		N/A	12	Others	Human topoisomerase I & II α	NC	0.039 & 0.126
MMV-390048		3.86	3.5	Kinase and phosphatase inhibitors	Pf PI4K	0.229	NC
Thimerosal		1.5	0.93	Antibiotics & antivirals	Unknown	NC	NC
Peruvoside		N/A	2.675	Cardiac glycosides	Human ATPase Na ⁺ /K ⁺ pump	9.95	0.02 & 0.032
Lanatoside A		N/A	0.86	Cardiac glycosides	Human ATPase Na ⁺ /K ⁺ pump	9.95	0.171 & 0.136
Givinostat hydrochloride		2.1	1.4	Anti-cancer: HDAC inhibitors	Human HDAC	NC	NC
CUDC-907		3.32	1.86	Anti-cancer: HDAC inhibitors	Human HDAC	0.018	0.03 & 0.003

MLN 576		N/A	5.02	Others	Human topoisomerase I & II	2.04	0.218 & 0.413
Convallatoxin		N/A	0.05	Cardiac glycosides	Human ATPase Na ⁺ /K ⁺ pump	0.016	0.026 & 0.033
Digitoxin		N/A	10.6	Cardiac glycosides	Human ATPase Na ⁺ /K ⁺ pump	NC	0.008 & 0.011
SkQ1		3.85	1.9	Others	Unknown	1.05	0.778 & 3.13
CHR-3996		0.07	7.23	Anti-cancer: HDAC inhibitors	Human HDAC	0.015	0.728 & 0.127
Cymarine		N/A	0.33	Cardiac glycosides	Human ATPase Na ⁺ /K ⁺ pump	9.95	0.071 & 0.102
Proscillaridin		0.3	0.08	Cardiac glycosides	Human ATPase Na ⁺ /K ⁺ pump	9.95	0.009 & 0.014
Alvespimycin hydrochloride		1.75	0.96	Others	Human HSP90	0.272	0.098 & 0.014
Halofuginone		0.24	4.3	Others	Human MMP-2	0.001	0.089 & 0.036
Octenidine		1.25	1.825	Antibiotics & antivirals	Unknown	NC	NC
Panobinostat lactate		0.3	0.22	Anti-cancer: HDAC inhibitors	Human HDAC	NC	NC

Mitoquinone mesylate		11.7	12.1	Others	Human mitochondria	NC	NC
Stilbazium iodide		8.745	1.57	Antibiotics & antivirals	Unknown	NC	NC
Zinc Pyrithione		1.675	2.6	Antibiotics & antivirals	Fungal proton pumps	NC	NC
Istaroxime		N/A	0.17	Cardiac glycosides	Human ATPase Na ⁺ /K ⁺ pump	NC	NC
Unidentified compound		N/A	0.04	Others	Unknown	NC	NC
(S)-(-)-Tetrahydro palmatine (L-THP)		N/A	0.001	Others	Unknown	NC	NC
Myristyl-gamma picolinium chloride		4.15	7.2	Antibiotics & antivirals	Unknown	NC	NC
Narasin		46	43	Antibiotics & antivirals	Dengue virus	NC	NC

Abbreviations:

N/A: not active in dose-response analysis

NC: not captured in primary screening

MAO: mono amino oxidase

HDAC: histone de-acetylase.

2. The authors should strengthen their description of the criteria (i.e. %inhibition in killing and/or retention activity) used for hitselection at the screening concentration which should be clearly stated. This reviewer could not readily find this information in the manuscript current form, and this must be provided to understand the rationale and statistical framework used by the authors to progress compounds to DRA and further characterization.

Hit selection criteria for primary screening are briefly described in the legend of the Fig.2 (lines 111-118) and more in detail at the Methods section (lines 475). To clarify the linear regression method, we added the following sentence: “*Only those compounds that fell outside the upper prediction band in both readouts were finally selected as hits*” The single 1.11 μM drug concentration for primary screening is indicated at the Methods section and now in the legend of Fig 1.

Hit confirmation criteria for dose-response analysis are described in the Results (lines 120-128) and Methods (lines 485-493). The DRA was just used to confirm the effect of hits selected by the primary screening, and broadly estimate their potency by determining their IC_{50} for killing and stiffening (Fig. 3, Table S1) but no IC_{50} cut-off value was used for further hit prioritization. Confirmation by DRA of stiffening, killing, or the coexistence of both was enough for hit prioritization. The following sentence was added in the line 121: “*119 hits were selected for dose-response analysis (DRA): three from the Pathogen Box, four from the Kinase Inhibitors Box, and 112 from the ReFrame library. Confirmation of the stiffening effect, killing effect or coexistence of both was the criterion for further analysis of hits. IC_{50} values were obtained for both killing effect and stiffening activity (Fig. 3, Table S1) but no IC_{50} cut-off value was used for further hit prioritization. Hits were deprioritized when no IC_{50} could be determined.*” The final selection of the 3 hits is described in the Table 1 and lines 166-168 of the main text. The following was added line 168: “*Briefly, selected hits that were not orally administered in animal models or in human subjects were excluded (44 confirmed hits excluded out of 76). Then, the remaining 32 hits were ranked for safety and PK. As anti-transmission imposes almost perfect safety, drugs that showed serious adverse events were excluded. Finally, hits with the best therapeutic window (serum peak greater or close to IC_{50}) were explored further. For example, MMV-390048, an antimalarial drug with a good safety profile, was excluded from further analysis because of an absent or very narrow therapeutic window (IC_{50} 3.5-3.9 μM , serum peak concentration 2.8 μM^{37}).*”

3. NFID609 (aka KAE609 and cipargamin) anti-gametocidal activity and effect on the rheology of the RBC has been previously reported and thus KAE609 was proposed to be a potent transmission blocking antimalarial drug candidate. It is thus very reassuring to see this compound coming out as a top hit of the primary screen. The authors have very interestingly decoupled the killing activity of the compound from the retention activity in the microfiltration device which should allow to dissect these two distinct pharmacological activities especially when we consider the lack of killing activity of TD-6450 (see below point). Have the authors considered testing some of the KAE609 drug resistant mutants in their assays? Would they expect to see an impact of the drug resistance mutations solely on killing activity or also in retention activity like reported in Zhang et al. 2016?

The potential impact of parasite resistance to ATP4 inhibitors on their transmission-blocking potential is indeed of major interest. Perhaps simplistically, we would consider killing as the ultimate consequence of drug-induced cell swelling, that induces gametocyte retention. If this assumption is correct, resistance should impact both effects. May we remind however, that any new determination of a “stiffening” IC_{50} on mature gametocytes requires at least 3 successful rounds of (partially unpredictable) sexual induction which takes 2 to 3 weeks, followed by a 10 to 12-day maturation period, each with several milliliters of packed RBC at very low hematocrit (many large volume flasks). Each deformability experiment is a tour-de-force requiring constant control of temperature (that must be tightly maintained at 37°C) during exposure, preparation, filtration and counts. Technical mistakes inevitably occur in such a complex process, each destroying 1 month of work. Determination of an IC_{50} on asexual parasites is,

literally, 10-100 times easier. In other words, any robust new determination of an IC50 can hardly be performed in less than 4-6 months (generating the whole set of results presented in this manuscript took more than 5 years). We hope that the reviewer will agree that testing the biomechanics of NITD609 drug-resistant mutants, a fascinating research trail, falls beyond the scope of this report.

4. The justification for the selection of only TD-6450 for more in-depth pharmacological analysis is a little perfunctory and few thoughts are given on how the data reported here for cipargamin may inform its further clinical development. The authors should consider running for cipargamin an analysis similar than that of TD-6450 since all the PK/PD phase I and phase IIa/b data are reported and available for cipargamin.

Cipargamin showed a stiffening IC50 of 150 nM where Cmax in humans is 630 nM at the lowest dose. The therapeutic window is relatively wide. These observations lead us to assign the highest score for PK to cipargamin and to skip a PK modelling step. Furthermore, cipargamin is at advanced stage of clinical development to treat malaria attacks. We shared our results with Novartis, who expressed interest but indicated, that, at the moment, development decisions were determined by ongoing Phase 3 trials. Studies on transmission-blocking would come after the NDA is obtained, an apparently usual (and probably wise) step-by-step process. Specifically designed studies will probably be needed as mature gametocytes often circulate days after curative treatment of the attack, or at unpredictable times in asymptomatic subjects. While the potential of ATP4 inhibitors as transmission-blockers is impressive, their effect is brief and exposed to a high risk of drug resistance, hence our strong motivation to keep exploring TD-6450, a probably weaker but potentially more sustainable (and possibly more resistant to resistance) transmission-blocker than cipargamin.

5. TD-6450 antimalarial property is largely unexpected given the origin of the compound as an antiviral drug with a target not expected to be present in Plasmodium. Its transmission blocking activity seems to be primarily driven by the activity on the iRBC deformability. The authors should clarify whether the transmission blocking activity of the compound is related solely to the retention activity given the moderate activity reported on the asexual stages of the parasite. Have the authors measured retention activity of TD-6450 in uninfected RBC? If the target of the TD-6450 transmission blocking activity in the RBC itself? This would offer some opportunity since one would expect a very high barrier to drug resistance but also some challenges as one might expect some on target toxicology. This is an important point that merits some mention at least in the discussion.

The reviewer is right: all along the campaign we have measured the **relative retention** of RBC containing mature gametocytes compared to that of uninfected RBC from the same culture. This approach was indeed unable to observe a potential direct effect of drugs on uninfected RBC. To answer this question, we performed a specific experiment and measured the retention of uninfected RBCs exposed to TD-6450 and NITD609. We tested both freshly collected RBC (to mimic what will occur in treated patients), or RBC maintained in culture conditions for 2 weeks (to mimic experimental conditions during screening and post-screening). The experiment was conducted with a negative control (DMSO-exposed RBCs). The results of this experiment are shown below.

Figure legend: Retention of uninfected RBC exposed to DMSO, TD-6450 or NITD609 during 24 hours before microfiltration. RBC were either used less than 72 hours after collection ("Fresh") or following 2 weeks in culture conditions at 37°C with medium change every other day ("Sham Cultured"). Positive values correspond to retention and negative values to enrichment of the RBC population of interest following filtration. Statistical analysis with one-way Anova test (**and *** $p < 0.01$, and < 0.001).

We observed no effect of TD-6450 or NITD609 on fresh RBC, in keeping with the absence of hematological adverse events with either drug during clinical trials. There was a 3-5% **enrichment** after exposure to both drugs, which may deserve specific explorations for other purposes (eg, prevention of damage on RBC during pre-transfusion storage). However, selective or not on uninfected RBC, this effect was too mild to have markedly modified the observed retention of gametocytes during the screening/post-screening campaign. Not least, when drugs will be used in patients, they will act on RBC harboring gametocytes, and on fresh RBC which are not affected by the drugs. A drug action predominant on the host RBC is not supported but this new set of data. That TD-6450 inhibits parasite growth at micromolar range concentration also suggests that its action is parasite-specific. This is reassuring for safety. Regarding the risk of drug resistance, it should be reduced by the proposed drug combination (NITD609+TD-6450) and by the relatively low number of gametocytes in circulation.

We have added the following sentence to the discussion (line 361): *That TD-6450 inhibits parasite growth at micromolar range concentrations suggests that its action is predominantly parasite-specific rather than based on a putative effect on uninfected RBC. Uninfected RBC*

were indeed not (or only mildly) affected by exposure to the TD-6450 or NITD609 in vitro (not shown).

6. The moderate activity of TD-6450 in the asexual stages assay offered the opportunity to conduct some drug resistance studies that yielded some reagents to further explore whether the effect on retention of the iRBC is truly decoupled from the antimalarial activity. Have the authors considered running microfiltration experiments with TD-6450 drug resistant mutants? Have the authors sequenced the genomes of those mutants? Have they identified significant single nucleotide polymorphisms? Please clarify.

The status of this part of our approach is less advanced than assumed by the reviewer. Whether the mild and unstable shift in “killing” IC50 observed for asexual stages corresponds or not to true resistance is not clear. Of note, IC50 increased after 5 pulses but went partially back to pre-pulses values after 10 pulses (Fig 5C). Experts disagree on the amplitude of an IC50 shift that definitely indicates phenotypic resistance. Our conclusion at the moment is that no drastic phenotype has emerged after 10 discontinuous exposures (pulses) although a 2-fold shift sometimes corresponds to the selection of mutants. Based on advises from specialists of parasite resistance to antimalarials (coauthors from Tres Cantos, Spain, and Pr. Benoit-Vical, University of Toulouse, France), our plan is to resume pressure until a statistically significant > 3-fold increase in IC50 is observed. When this is achieved, we will run the phenotypic tests and genome sequencing.

7. The authors have carefully repeated the experiments to ensure adequate representation of the variability of the data (Fig. 4, 5 and 7) but the reviewer was not able to determine whether the replicates were technical or biological and this should be clarified to the reader.

Replicates are both technical and biological. At Fig. 4 B-C and 7C: each dot is a well of a microfiltration plate and they are pooled together from 5 (4B), 4 (4C) and 3 (7C) experiments. For each experiment, 8 wells per condition were loaded. The replicates are then technical considering a single experiment, but also biological when we consider different experiment, each of them with a specific gametocyte induction (biological difference) and a different microfiltration plate (again technical). Fig. 5 B-C: Experiments were done in triplicate, with only technical replicates here. The legend of the Figures 4 and 7 were amended, as follows:

Lines 221:

(C) Cumulative dot-plot of 4 microfiltration experiments where gametocytes were exposed to the drugs for 24 hours and kept in culture for an additional 24 hours after removing the drug. Dots indicate the retention rate of single wells of 96-well microfiltration plates. For Panels B & C, each experiment was performed with a single gametocyte induction and a single 8-well column for each condition was loaded, unless the cultured gametocyte population was not large enough to fill the entire column. Numbers of repetitions are, from left to right: 40 (8 for each experiment), 40 (8 for each experiment), 32 (8 for each of 4 experiments), 32 (8 for each of 4 experiments).

Lines 285:

“Each experiment was performed with a single gametocyte induction and a single 8-well column for each condition was loaded, unless the cultured gametocyte population was not large enough to fill the entire column. Numbers of repetitions are, from left to right: 23 (8 for the first

2 experiments, 7 for the 3rd, 16 (8 for each of the first 2 experiments), 23 (8 for the first 2 experiments, 7 for the 3rd), 7 (3rd experiment only) and 7 (3rd experiment only).”

Minor

comments:

1. Fig 1. KDU731 is a known antimalarial drug candidate and a lipid kinase (PI4K) inhibitor and is not appropriately classified in drug families (ie it is not a phosphatase inhibitor). Please describe selection criteria and screening concentration on figure and/or legend.

In the revised Fig.1C (showed below) and in the text (line 161), the name of the group was modified from “Kinase phosphatase inhibitors” to “Kinase *and* phosphatase inhibitors” to avoid any misinterpretation. KDU731 is correctly included in this group as a kinase inhibitor. For each group, one hit was selected for illustration.

Fig.1. High-throughput screening based on mitochondrial staining and cell deformability identifies compounds with both killing effect and stiffening activity on *P. falciparum* late gametocytes. Screening progression cascade of three different libraries: Malaria Pathogen Box

(A), Kinase Inhibitors Box (B), and ReFrame library (C). (A) 3 hits from primary screening were submitted to dose-response analysis along with 12 compounds found active in some but not all screening replicates. The 3 hits were confirmed but none of them was selected for further post-screening validation. (B) 4 hits from primary screening along with 5 compounds found active in some but not all screening replicates were submitted to dose-response analysis raising 3 confirmed hits. None of them was selected for further post-screening validation. (C) 112 hits from primary screening were submitted to dose-response analysis, raising 74 confirmed hits. 63 compounds with uninterpretable results during primary screening were added to the hits for dose-response analysis raising additional 2 confirmed hits. The 76 confirmed hits were allocated to 7 groups (panels on the right), based on their activity and molecular target. For each group, one hit has been selected to represent it. Hit scoring based on route of administration, safety in human subjects, and pharmacokinetics resulted in the selection of 3 drugs submitted to final confirmation experiments (dark blue). Compounds from all libraries were screened at 1.11 μM .

2. Table 1 MMV390048 is also a PI4K inhibitor with a large volume of PK/PD available. Why did the authors deprioritize further analysis for this compound? Was this solely on PK data? This might be worth being more explicit about this given the interest for this drug target.

MMV390048 was deprioritized because the observed IC_{50} in our assay (3.9 μM and 3.5 μM for killing and stiffening, respectively) (Table S1) and the observed serum peak (2.8 μM when the highest 120 mg dose was administered, McCarthy et al 2020 Clin. Inf. Dis) do not anticipate a therapeutic window. The following sentence was added before the line 172: “*For example, MMV-390048, an antimalarial drug with a good safety profile, was excluded from further analysis because of an absent or very narrow therapeutic window (IC_{50} 3.5-3.9 μM , serum peak concentration 2.8 μM ³⁷.*”

3. Fig. 2. Why did compounds fail to reconfirm (blue empty triangles)? Was this after re-synthesis of the compounds or because the further biological replicates were conducted and failed?

Compounds for DRA were prepared from the same master plate and dispensed in new plates with serial dilution from 10 μM to 5 nM. 66% of hits (74 out of 112) were confirmed, the other were eliminated because no IC_{50} could be determined by DRA. We consider this result as a biological replicate that was conducted and failed.

4. Please clarify the meaning of “not interpretable” on line 128. No data or data that do not pass QC?

To better explain the source of the 63 not interpretable wells (among the 12,805 compounds of the ReFrame library), the text was changed in the lines 133 as follows:

“The results for 63 of the 12’805 compounds in the ReFrame library were not interpretable upon primary screening. For 15 compounds, the corresponding well of the 384-well microfiltration plate was not operational due to leakage of microspheres, a rare event²⁷. For 46 compounds, the microscope failed to capture at least one readable image. Finally, we tested sildenafil and tadalafil by DRA. Although these drugs were not captured by the primary screen they have been previously reported to induce stiffening of stage V gametocytes^{29,30}.”

And the following sentence was added in the line 450:

“The 384-well filter plates were then stored at -20°C until use. In some of the 384-well microsphere plates leakage of microspheres occurred in one or more wells. The wells with leakages were sealed. Plates displaying more than 5 sealed wells were not used.”

5. Line 129. Please explain the impact of the technical challenges of the assay on the overall screen. Do we expect high number of false positives or are we missing a lot of false negatives?

The answer to the previous point addresses this query. 63 technical deficiencies do not heavily impact the overall performance of the screen. The accurate capture of all known or suspected “actives” suggests there were only few false negatives. Supposedly, the DRA step eliminated most false positives. 66% of hits (74 out of 112) were confirmed, the other were eliminated because no IC50 could be determined by DRA.

Screening campaign captured drugs with known gametocyte killing effects, such as methylene blue, Pf-ATPase 4 inhibitors, and Pf-PI4K inhibitors, PA92, was captured by the screening campaign, confirming that these inhibitors are promising targets for transmission-blocking strategies

6. Line 175-176 please explain why the authors did not follow-up on L-THP?

L-THP was in one of the 63 not interpretable wells during screening campaign (failing imaging, see minor point 4), so only DRA results were available. Further analysis in Paris did not confirm the stiffening effect. We mentioned this compound because of its anti-plasmodial activity (Malebo et al. BMC Complementary and Alternative Medicine 2013) on asexual stages. The following was added line 187: *“L-THP was not explored further because, despite its known activity of asexual stages⁴³, its stiffening activity was not fully confirmed upon post-screening”*.

7. Table 2 could be considered as supplemental if data is available online in another format.

These data from the phase I trial will be published here for the first time and are not available on-line.

8. Line 187 has a typo (activities)

Corrected Thank you.

9. Line 199 Provide references for “known parasite-swelling”.

A new reference (number 44 Chavchich M, 2016) was added line 208: *“Microscopic observations confirmed the known parasite-swelling effect mediated by NITD609 while no morphological change was observed in gametocytes exposed to TD-6450⁴⁴.”*

10. Line 204-205 NITD609 is not preferentially active on male gametocytes.

The expression “predominantly active on male gametocytes” refers to TD-6450, not NITD609. *“These readout-dependent differences suggest that TD-6450 is predominantly active on male gametocytes...”*

11. Line 256-258 efforts should be made to clarify the rationale for the selection of a 200 nM threshold and consider the impact of this prediction on a potential novel drug combination drugs regimen. How many days would a patient need to take the drug to significantly impact malaria transmission? 1000 mg is a relatively large dose, and this could also be a consideration for a suitable drug partner for a combination?

The following sentence was added line 275: “*This concentration (200 nM) is the best compromise between entire population coverage (including all modelling covariates) and the in vitro stiffening activity of TD-6450 expected to translate in a marked in vivo clearance.*”

When a single 1000 mg dose is administered with food, all members of population have at least 200 nM of TD-6450 in serum for 8 hours (Fig. 7B) a concentration at which TD-6450 is active in the spleen-mimetic device (Fig. 7C). This peak concentration is still 2.5 times lower than what has been achieved after multiple doses, without any safety signal when the drug was administered for 12 days. The safety margin is therefore still very wide. Whether the best transmission-blocking strategy will rely on a single dose mass drug administration or, for example, repeated administration at the beginning of transmission period cannot be determined at this time. In an ambitious malaria control program, taking one (or two) pill(s) per week is very feasible and will complement additional approaches (bednets, IRS, etc.).

12. Efforts should be made to make the discussion more concise (lots of repetitions of the results) and discuss more transparently the opportunity and challenges of the proposed approach to block malaria transmission.

In line with the reviewer’s advice, we have shortened the discussion to 1300 words, despite 6 additions justified by reviewers comments.

We have removed the following paragraphs

~~Any one approach to malaria elimination in highly endemic African countries will most likely prove insufficient⁴. Old and new contributions^{63,64} shape a realistic model where a combination of partially effective tools drives the situation toward progressively increasing control. A combination of vaccination to protect a part of the population and bed nets to lessen human-vector contacts may be advantageously complemented by a drug to reduce the proportion of infected vectors. Transmission blocking agents do not need to be as rapidly effective as antimalarial drugs used to cure acute, potentially severe malaria attacks. However, rapid transmission blocking may be useful in the context of epidemics.^{4,65}~~

AND

~~The spleen mimetic approach used here, focused on parasite deformability rather than on parasite killing, is highly innovative. TD-6450 and several other effective compounds had indeed not been captured by previous screening campaigns. The stiffening activity of TD-6450 on mature gametocytes persists for at least 24 hours after drug washout. An even longer post-exposure activity may exist, but could not be explored *in vitro* due to the exquisite fragility of mature gametocytes. The prolonged stiffening activity of TD-6450 and its long half life are expected to enhance the phagocytosis of spleen-retained gametocytes, thereby reducing the risk of their reappearance in the circulation once the drug clears.⁴⁶~~

. To meet the reviewer's expectation to “**discuss more transparently the opportunity and challenges of the proposed approach to block malaria transmission**”, we have nuanced the following statements: “*Therefore, a drug or a drug combination inducing the clearance of 70–80% of gametocytes may impact transmission*”, and “*A decrease in gametocytemia is indeed directly related to reduced transmission, and we provide robust results suggesting that TD-6450 and NITD609 may reduce gametocytemia*”, and we have added the following paragraph at the end of the discussion: *Developing drugs to block the transmission of malaria is confronted to ethical, pharmaceutical, and logistical challenges. If the potential transmission-blocking effect of NITD609 and TD-6450 is confirmed in a clinical trial, adding these safe drugs to the armamentarium may contribute to the momentum of an original approach aiming at a sustained control of malaria.*” We hope that the reviewer will agree that discussing in details all potential challenges would take us back to a less concise discussion.

13. Line 335 the word “enzymes” is a poor generic descriptor of PfATP4 (consider channels or ion transporters).

Line 335 (359 in the new version of the manuscript) the word “enzymes” was replaced with “*inhibitors*”. We also corrected the number of drugs in each group (correct in Table S1 but inexact for one compound in Fig 1 and text).

List of references ordered by alphabetical order:

- Andrews, K. T. *et al.* Towards histone deacetylase inhibitors as new antimalarial drugs. *Curr. Pharm. Des.* **18**, 3467–3479 (2012)
- Buffet, P. A. *et al.* Ex vivo perfusion of human spleens maintains clearing and processing functions. *Blood* **107**, 3745–3752 (2006)
- Buffet PA, *et al.* The pathogenesis of Plasmodium falciparum malaria in humans: insights from splenic physiology. *Blood.* **117**(2):381-92 (2011)
- Churcher TS, *et al.* The impact of pyrethroid resistance on the efficacy and effectiveness of bednets for malaria control in Africa. *Elife.* **5**:e16090 (2016)
- Cranston, H. A. *et al.* Plasmodium falciparum maturation abolishes physiologic red cell deformability. *Science* **223**, 400–403 (1984).
- Duffier Y, *et al.* A humanized mouse model for sequestration of Plasmodium falciparum sexual stages and in vivo evaluation of gametocytidal drugs. *Sci Rep.* **6**:35025 (2016)
- Henry B, *et al.* The human spleen in malaria: filter or shelter? *Trends Parasitol.* **36**(5):435-446 (2020)
- Jandl JH, *et al.* The destruction of red cells by antibodies in man. I. Observations of the sequestration and lysis of red cells altered by immune mechanisms. *J Clin Invest.* **36**(10):1428-59 (1957)

- Kho S, *et al.* Hidden Biomass of Intact Malaria Parasites in the Human Spleen. *N Engl J Med.* **384**(21):2067-2069 (2021)
- Kho S, *et al.* Evaluation of splenic accumulation and colocalization of immature reticulocytes and Plasmodium vivax in asymptomatic malaria: A prospective human splenectomy study. *PLoS Med.* **18**(5):e1003632 (2021)
- Malebo, H. M. *et al.* Anti-protozoal activity of aporphine and protoberberine alkaloids from Annickia kummeriae (Engl. & Diels) Setten & Maas (Annonaceae). *BMC Complement. Altern. Med.* **13**, 48 (2013).
- McCarthy, J. S. *et al.* A Phase 1, Placebo-controlled, Randomized, Single Ascending Dose Study and a Volunteer Infection Study to Characterize the Safety, Pharmacokinetics, and Antimalarial Activity of the Plasmodium Phosphatidylinositol 4-Kinase Inhibitor MMV390048. *Clin. Infect. Dis. Off. Publ. Infect. Dis. Soc. Am.* **71**, e657–e664 (2020)
- Meibalan, E. *et al.* Plasmodium falciparum Gametocyte Density and Infectivity in Peripheral Blood and Skin Tissue of Naturally Infected Parasite Carriers in Burkina Faso. *J. Infect. Dis.* **223**, 1822–1830 (2021)
- Rottmann, M. *et al.* Spiroindolones, a Potent Compound Class for the Treatment of Malaria. *Science* **329**, 1175–1180 (2010)
- Roussel, C. *et al.* Rapid clearance of storage-induced microerythrocytes alters transfusion recovery. *Blood* **137**, 2285–2298 (2021)
- Safeukui, I. *et al.* Retention of Plasmodium falciparum ring-infected erythrocytes in the slow, open microcirculation of the human spleen. *Blood* **112**, 2520–2528 (2008).
- Safeukui I, *et al.* Surface area loss and increased sphericity account for the splenic entrapment of subpopulations of Plasmodium falciparum ring-infected erythrocytes. *PLoS One.* **8**(3):e60150 (2013)
- Stepniewska K, *et al.* Efficacy of Single-Dose Primaquine With Artemisinin Combination Therapy on Plasmodium falciparum Gametocytes and Transmission: An Individual Patient Meta-Analysis. *J Infect Dis.* **225**(7):1215-1226 (2022)
- Tibúrcio M, *et al.* A switch in infected erythrocyte deformability at the maturation and blood circulation of Plasmodium falciparum transmission stages. *Blood.* **119**(24):e172-80 (2012)

REVIEWERS' COMMENTS

Reviewer #1 (Remarks to the Author):

Thank you to the authors for addressing my questions and concerns with the revised manuscript. I support the publication of the manuscript as submitted.

There is a typo on line 83: theme should be them

Reviewer #2 (Remarks to the Author):

No further comments, other than to thank the authors for so comprehensively responding to the reviews. It was already an excellent and exciting paper - I believe that the modifications and added details have helped clarify the findings and impact even further.

Reviewer #3 (Remarks to the Author):

Thanks to the reviewers for addressing constructively all reviewers' comments. They have adequately answered my questions in their rebuttal letter, and I recommend publication of this manuscript.